# Sparse Covariance Supervised Principal Component Analysis

## Abstract

Principal component analysis (PCA) is one of the most well-studied machine learning methods of the last century. However, the principal components derived by PCA are not guaranteed to be response-informative and are usually dense, meaning they are hard to interpret in high-dimensional settings. The former has led to the development of supervised PCA techniques where the response is usually incorporated in an objective function to guide informative projections and enhance predictive accuracy, while the latter to sparse PCA methods that seek to induce sparsity by shrinking non-significant variables to zero, effectively improving interpretability. Sparse supervised PCA methods seek to combine the two concepts as a means of simultaneous supervised dimensionality reduction and variable selection, but they usually depend on iteratively biconvex solutions of auxiliary objective functions, with no robust convergence guarantees and are sensitive to initialisation. We propose a novel sparse covariance supervised PCA (SCS-PCA), which seeks to trade-off prediction accuracy and sparsity performance. We impose an $\ell_1$ penalty on a supervised objective function and we employ manifold proximal gradient descent to solve the derived optimisation problem, which guarantees global convergence to a stationary point. Numerical results from simulations and real-world microarray data illustrate that SCS-PCA provides competitive performance in prediction tasks, selects fewer features, and outperforms all projection-based methods in variable selection accuracy.

## 1 Introduction

Principal component analysis (PCA) (Jolliffe, 2002) is one of the most popular data mining techniques used for dimensionality reduction, feature extraction, data visualisation etc., with applications in numerous fields such as genetics and bioinformatics (Elhaik, 2022), computer vision (Mehrabinezhad et al., 2024), chemometrics (Bin et al., 2013) and finance (Mavungu, 2023), to name a few. It projects the original high-dimensional data into a lower dimensional, orthogonal subspace by considering linear combinations of the original data that preserve as much of the intrinsic variability as possible while also being uncorrelated (Jolliffe & Cadima, 2016).

By definition, PCA has two notable limitations. First, PCA is an unsupervised learning method and is not guaranteed to provide informative projections with respect to the response variables, when the latter are available, leading to poor predictive performance in subsequent prediction models (Ritchie et al., 2019). The second shortcoming is that the loadings (coefficients) of the principal components are typically dense, meaning most or all original features contribute to each principal component, hindering interpretability, especially in high-dimensional settings, where the number of features ranges from thousands to millions (Zou et al., 2006).

The former limitation has motivated the development of supervised PCA dimensionality reduction techniques, that seek to optimise a supervised objective function similar to that of PCA. Bair & Tibshirani (2004) and Bair et al. (2006) proposed a two-stage supervised PCA, incorporating the response during the first stage, where its correlation to the data is measured in order to define a subset of significant features. The second stage applies a standard PCA on the created subset. However, this method does not integrate the response directly during learning. Barshan et al. (2011) addressed this issue by considering a single objective function that uses the Hilbert-Schmidt independence criterion (HSIC) to maximise the dependence

between kernels in the reproducing kernel Hilbert space (RKHS). Both methods, however, oversee PCA's main aim which is maximising the variance explained by the projected data. Papazoglou & Yin (2025) recently proposed an ensemble method that seeks to trade-off maximising the covariance of the response and the data and the variance of the projected data. Since the two objectives are competing, they are balanced via a tunable regularisation parameter, improving interpretability and prediction accuracy. Several other supervised PCA methods have been proposed recently, e.g., see Ritchie et al. (2019), Ritchie et al. (2020), Pascual & Yee (2022) with a good overview given in Chao et al. (2019) and Xu et al. (2021).

The latter limitation is also well known and explored thoroughly in the literature, motivating the development of sparse PCA methods. The introduction of sparsity aids in improving interpretability by reducing the number of non-significant features in a dataset. For example, in DNA microarray experiments, gene expression data often consist of measurements from tens of thousands of genes, yet only a small fraction are biologically relevant for accurate classification of phenotypes for diagnosing a disease (Zhang & Deng, 2007). Principal components from PCA are linear combinations of nearly all genes, making it difficult to interpret and identify a subset of significant biomarkers. A closely related situation arises in statistical genetics, where fine-mapping analyses seek to isolate the few putatively causal variants driving an association signal from many correlated variants in linkage disequilibrium at a locus (Wu et al., 2026). In both cases, the scientific objective is itself one of variable selection, since a representation that retains only a handful of genes or variants is far more actionable than a dense combination of thousands. By incorporating sparsity, either in the form of a penalty (e.g. $\ell_1$) or a constraint (e.g. $\|\cdot\|_1 \leq \eta$), sparse PCA shrinks loadings of irrelevant genes to zero (Zou et al., 2006), effectively performing variable selection in addition to dimensionality reduction. The sparse principal components derived from sparse PCA are linear combinations of only a small subset of genes, enhancing interpretability and potentially improving prediction accuracy. In the context of gene expression analysis, sparse PCA isolates the genes that contribute most to explaining the variance, thereby improving biological insight and facilitating the identification of key biomarkers. Many different versions of sparse PCA have been proposed over the years with a good overview found in Zou & Xue (2018) and Guerra-Urzola et al. (2021).

The manner in which sparsity is induced also matters, and not every norm is suited to variable selection. Ridge ($\ell_2$) regularisation, for instance, shrinks the loadings towards zero and improves conditioning, but it does not achieve exact zeros; every variable is retained, so it performs no selection and does not reduce the number of features that must be interpreted. The ideal $\ell_0$ penalty, which directly counts the number of selected variables, does induce selection but renders the problem combinatorial and intractable. The $\ell_1$ norm is the tightest convex surrogate and, crucially, admits a closed-form proximal (soft-thresholding) operator, which makes it both the standard choice in the literature and the one most compatible with the manifold proximal optimisation we employ. The most notable sparse PCA formulation, proposed by Zou et al. (2006), employs an elastic net penalty, i.e. a combination of $\ell_1$ and $\ell_2$ penalties.

Sparse methods are not limited to PCA, with similar extensions appearing, for example, in partial least squares (PLS) and linear discriminant analysis (LDA), namely sparse PLS (Chun & Keleş, 2010) and sparse LDA (Clemmensen et al., 2011) respectively. Most relevant to our work is the inclusion of sparsity into supervised PCA. Sharifzadeh et al. (2017) proposed supervised sparse PCA (SSPCA), where an $\ell_1$ norm constraint was imposed in the projection matrix of supervised PCA using HSIC (Barshan et al., 2011) to derive sparse supervised principal components using penalised matrix decomposition (PMD), while Feng et al. (2019) proposed supervised discriminant sparse PCA (SDSPCA), where discriminative information and sparsity are introduced directly into PCA, through an $\ell_{2,1}$ penalty directly on the principal components, for tumor classification. The latter was further extended in Shi et al. (2020) to incorporate projected clustering with adaptive neighbours into SDSPCA to form SDSPCA with adaptive neighbours (SDSPCAAN).

SDSPCA and SDSPCAAN induce sparsity directly on the principal components rather than the loading matrix and hence cannot be used for variable selection, unlike SSPCA. Additionally, they can only be applied for binary classification tasks while SSPCA can also be applied in regression or multi-label tasks. SSPCA, on the other hand, does not address the trade-off between sparsity and prediction accuracy directly since sparsity is incorporated as a constraint rather than as a penalty and also requires careful specification of a kernel for the response variables. A common feature among all three methods is the optimisation scheme employed to solve the respective problem. Specifically, all sparse supervised PCA optimisation problems are

non-convex and non-smooth due to the orthogonality and sparsity constraints. As a result, sparse supervised PCA methods employ auxiliary objective functions to translate the original non-convex optimisation problem into a surrogate biconvex problem that can be solved using iteratively alternating algorithms. Feng et al. (2019) and Shi et al. (2020) used an iteratively alternate algorithm to solve SDSPCA and SDSPCAAN respectively, while Sharifzadeh et al. (2017) solved SSPCA using PMD which has also been used to solve the sparse PCA problem (Witten et al., 2009) and is likewise based on an alternating algorithm. While these schemes are simple to implement and some are accompanied by convergence analyses (Feng et al., 2019), the guarantees are of a weaker type, as they concern the alternating iteration or a surrogate formulation, are sensitive to initialisation, and do not establish global convergence to a stationary point of the original penalised problem. Here, *global convergence* refers to convergence to a stationary point from an arbitrary initialisation, which is frequently used in manifold theory (Boumal et al., 2019), and not a global minimiser, which is generally unattainable for a non-convex problem.

The difficulty in obtaining rigorous optimisation schemes for sparse methods stems from the non-convexity of the feasible set, i.e. the orthogonality constraint, rather than the objective itself. When the objective can be decomposed into a smooth term and a convex, possibly non-smooth term, the proximal gradient method (Combettes & Wajs, 2005) is applicable to the resulting non-smooth optimisation problem. However, proximal gradient descent is defined in Euclidean settings, while the optimisation problem we consider lies on the Stiefel manifold, i.e. the set of all orthonormal $p$-frames in $\mathbb{R}^n$. Recent developments in manifold optimisation have transferred the method from the Euclidean setting to manifolds while preserving its favourable properties. For example, the Riemannian proximal gradient method of Huang & Wei (2022) was used to solve sparse PCA directly, without heuristic approximations, and was shown to converge globally to a stationary point under minimal requirements. For the formulation presented in this paper we adopt the closely related manifold proximal gradient (ManPG) algorithm of Chen et al. (2020), which is specifically designed for non-smooth optimisation over the Stiefel manifold and, when applied to our objective, converges globally to a stationary point. Although ManPG itself was introduced for sparse PCA, applying to the supervised covariance formulation underlying our proposed method yields, to the best of our knowledge, the first sparse supervised PCA method that directly optimises its exact $\ell_1$-penalised objective with global convergence guarantee to a stationary point, in place of the surrogate formulations that employ initialisation-dependent alternating algorithms.

We propose a novel formulation for supervised sparse PCA that seeks to perform simultaneous supervised dimensionality reduction and variable selection. The derived non-smooth optimisation problem is solved using manifold proximal gradient descent to establish global convergence to a stationary point. Our proposed method seeks to (i) derive interpretable supervised sparse projections, (ii) establish global convergence to a stationary point for sparse supervised PCA through a robust optimisation scheme, (iii) identify a subset of relevant features through supervised dimensionality reduction and (iv) improve prediction performance for subsequent prediction models.

## 2 Background

### 2.1 PCA and Sparse PCA

Throughout the paper, unless stated otherwise, we assume the following notation. We denote $X \in \mathbb{R}^{n \times p}$ the feature data matrix and $Y$ as the $n \times k$ matrix containing the response variables. We assume that both matrices have already been centred.

PCA learns an orthogonal projection of the original data into a lower dimensional linear subspace, i.e. $Z = XW$, such that the variance of the projected data is maximised, or equivalently, the reconstruction error is minimised,

$$\min_{W:W^\top W=I_q} \|X - XWW^\top\|_{\mathcal{F}}^2,$$

where $W \in \mathbb{R}^{p \times q}$ the projection matrix satisfying the orthonormality condition, $W^\top W = I_q$. The solution to PCA can be derived either by the eigenvalue decomposition of the covariance matrix $\Sigma = X^\top X$, or via singular value decomposition (SVD) directly on $X$. The new projections are linear combinations of the

variables in the original dataset that are uncorrelated with each other (Jolliffe & Cadima, 2016). However, in (ultra-)high-dimensional settings, this can decrease interpretability significantly.

Sparse PCA (SPCA) improves interpretability by enforcing sparsity on the loadings, removing non-significant variables. Sparsity is usually integrated in the PCA framework in the form of an $\ell_1$ penalty or constraint (Tibshirani, 1996), although other penalties can be used as well, e.g. $\ell_0$, elastic net etc. The most notable SPCA method was proposed in Zou et al. (2006), where an elastic net penalty was utilised to produce modified principal components with sparse loadings, while maintaining computational efficiency and scalability. Zou et al. (2006) used a regression formulation for SPCA,

$$\left(\hat{P}, \hat{W}\right) := \arg\min_{P,W} \|X - XWP^\top\|_{\mathcal{F}}^2 + \lambda \sum_{j=1}^{k} \|w_j\|^2 + \sum_{j=1}^{k} \lambda_{1,j} \|w_j\|_1,$$

where $P, W \in \mathbb{R}^{p \times k}$ are matrices of loadings, subject to $P^\top P = I_k$, $w_j$ is the $j$-th column of $W$, $\lambda > 0$ is the ridge penalty and $\lambda_{1,j}$ controls sparsity for the $j$-th principal component. The solution is computed via an alternating algorithm, optimizing $W$ and $P$ iteratively.

## 2.2 Supervised PCA Methods

Numerous supervised PCA extensions have been proposed over the last two decades, with comprehensive reviews in Ghojogh & Crowley (2019) and Xu et al. (2021). The first such method, introduced in Bair & Tibshirani (2004) and Bair et al. (2006), employs a two-stage approach: selecting features correlated with the response via a threshold rule, followed by standard PCA. However, this method fails to integrate the response during learning of the projection, potentially yielding less informative projections (Ritchie et al., 2019).

Barshan et al. (2011) addressed this issue using the Hilbert-Schmidt independence criterion (HSIC) to measure the dependence between the data and the response variables in reproducing kernel Hilbert spaces (RKHS). They incorporate the labels directly during the learning stage using the following optimisation problem,

$$\max_{W: W^\top W = I_q} \text{tr}(W^\top X^\top K X W) \tag{1}$$

where $K = K(Y)$ is a kernel of $Y$, e.g. radial basis function (RBF) kernel for regression. The solution to (1) is derived, similar to PCA, by considering the eigenvalue decomposition of the matrix $Q = XKX^\top$.

Recently, Papazoglou & Yin (2025) proposed covariance supervised PCA (CSPCA), which balances supervised relevance and interpretability by maximizing the covariance of the projected data with the response variables while also preserving the variability of the data,

$$\max_{W: W^\top W = I_q} \text{tr}(W^\top C W), \tag{2}$$

where $C = X^\top Y Y^\top X + \kappa X^\top X$ combines the supervised and unsupervised objectives via a tunable hyperparameter $\kappa > 0$. The solution derives from the top-$q$ eigenvectors of $C$. For classification tasks, the use of a delta kernel is recommended,

$$\delta(y, y') = \begin{cases} 1, & \text{if } y = y' \\ 0, & \text{if } y \neq y', \end{cases}$$

instead of $YY^\top$ to align the binary nature of the response with the model and also allow for non-linear relationships between the data and the response. CSPCA maintains data covariance structure, approximately matches PCA in explained variance, and is competitive to other supervised PCA methods in prediction accuracy for both regression and classification tasks.

## 2.3 Sparse Supervised PCA Methods

The method proposed by Barshan et al. (2011) still faces the issue of generating dense supervised principal components which consist of all the original variables and hence are difficult to interpret. Using the formulation in (1), Sharifzadeh et al. (2017) imposed an $\ell_1$ norm constraint on the eigenvectors of $W$ in (1) to

induce sparsity and remove non-significant variables by defining sparse supervised PCA (SSPCA),

$$\max_{W} \operatorname{tr}(W^\top X K X^\top W), \quad \text{such that} \quad \|W\|_1 \leq c, \tag{3}$$

where $1 \leq c \leq \sqrt{p}$ controls the sparsity. SSPCA preserves only the most relevant features improving interpretability and enhancing generalisation in high-dimensional settings. To solve SSPCA, Sharifzadeh et al. (2017) used the penalised matrix decomposition (PMD) method by Witten et al. (2009) that has also been used in sparse PCA applications. SSPCA applies sparsity on the loadings and can be applied to both regression and classification settings, while also effectively performing variable selection. However, SSPCA does not solve (3) directly, instead it employs a number of approximation steps, reformulating the original objective function. It also requires careful specification of the kernel applied to the response, which can affect significantly the performance of the method. For the experiments conducted in this paper, we consider the RBF kernel for regression tasks and the delta kernel for classification tasks.

Feng et al. (2019) proposed supervised discriminative sparse PCA (SDSPCA), a sparse supervised PCA method that incorporates class labels on the PCA problem, combining discriminative information and sparsity. The major difference from existing methods is that sparsity is applied directly on the principal components instead of the loadings. Mathematically, Feng et al. (2019) defined SDSPCA as

$$\min_{W,Q,A} \|X^\top - WQ^\top\|_{\mathcal{F}}^2 + \alpha\|Y - AQ^\top\|_{\mathcal{F}}^2 + \beta\|Q\|_{2,1}, \quad \text{subject to} \quad Q^\top Q = I, \tag{4}$$

where $Q$ represents the sparse principal components, $\alpha$ and $\beta$ are scale weights, $A$ is a linear transformation matrix that maps the sparse principal components ($Q$) to the class labels ($Y$) and $\|\cdot\|_{2,1}$ corresponds to the $\ell_{2,1}$ norm. From (4) we can observe how sparsity ($\beta$) is imposed on the principal components rather than the loadings as is the case for SSPCA in (3). As a result, SDSPCA is not suitable for variable selection since we cannot identify non-significant features, rather principal components only. Additionally, Feng et al. (2019) did not solve (4) directly but instead defined an auxiliary objective function to optimise using $\ell_{2,1}$ norm optimisation. The solution is then derived using an iterative algorithm that alternates between updating the loading matrix $W$, the transformation matrix $A$, the sparse principal components $Q$ and a diagonal matrix $V$ introduced in the auxiliary objective, complicating interpretation. In contrast to SPCA, SSPCA and the method we herein propose, the loading matrix derived from SDSPCA will not have sparse entries, since sparsity appears directly on the principal components in $Q$. SDSPCA can be applied only for classification settings, limiting its application spectrum. This method has further been extended to incorporate neighbour information during learning (Shi et al., 2020), but we stick to the original formulation by Feng et al. (2019).

### 2.4 PLS and Sparse PLS

PLS regression seeks a set of latent vectors that performs a simultaneous decomposition of the data and the response variables such that the covariance is maximised. A regression step is then performed to predict $Y$ using the decomposition of $X$. Mathematically, PLS can be defined as

$$X = PU_x^\top + E_x,$$
$$Y = KU_y^\top + E_y,$$

where $P, K \in \mathbb{R}^{n \times q}$, for the $q$ extracted components known as latent vectors, $U_x \in \mathbb{R}^{p \times q}$ and $U_y \in \mathbb{R}^{k \times q}$ are matrices of coefficients (loadings) and $E_x \in \mathbb{R}^{n \times p}$ and $E_y \in \mathbb{R}^{n \times k}$ are matrices of random errors. The components are extracted sequentially via deflation. The first column of $U_x$ is obtained as the leading left singular vector of $X^\top Y$, and the corresponding latent vector is the first column of $P$, given by $X u_{x,1}$. After computing the corresponding loadings, both $X$ and $Y$ are deflated by removing the contribution of the first latent vector, and the procedure is repeated on the residuals to extract subsequent components.

Sparse PLS (SPLS), proposed by Chun & Keleş (2010), extends PLS by imposing an $L_1$ constraint on the objective function of PLS. To formulate the SPLS optimisation problem, Chun & Keleş (2010) generalised the regression formulation used in SPCA by Zou et al. (2006), imposing the $L_1$ penalty onto a surrogate of the direction vector instead of the original direction vector, i.e.

$$\min_{\alpha,\gamma} -\kappa\alpha^\top M\alpha + (1-\kappa)(\gamma-\alpha)^\top M(\gamma-\alpha) + \lambda_1|\gamma_1| + \lambda_2|\gamma_2| \quad \text{s.t.} \quad \alpha^\top\alpha = 1,$$

where $M = X^\top Y Y^\top X$, $\lambda_1$ encourages sparsity on $\gamma_1$ and $\lambda_2$ accounts for potential singularity in $M$. The solution to (5) is derived by alternatively iterating between solving for $\alpha$ for $\gamma$ fixed and solving for $\gamma$ for $\alpha$ fixed. As in standard PLS, multiple sparse components are extracted sequentially: after obtaining the first sparse direction and the corresponding latent vector, both $X$ and $Y$ are deflated and the optimisation is repeated on the residual matrices to obtain the next component. This process is iterated until the desired $q$ components have been extracted.

## 3   Preliminaries

We first provide some key definitions. Throughout the PCA literature and its supervised extensions, the Frobenius norm is the most used norm as it is an essential part of the objective function of each method.

**Definition 1** (Frobenius Norm). *Assume an $n \times p$ real matrix $Z$. The Frobenius norm of $Z$ is defined as the square root of the sum of the absolute squares of its elements,*

$$\|Z\|_{\mathcal{F}} = \sqrt{\sum_{i=1}^{n} \sum_{j=1}^{p} |z_{ij}|^2} = \sqrt{\mathrm{tr}(Z^\top Z)},$$

*where $\mathrm{tr}(\cdot)$ denotes the matrix trace operation.*

In the example of PCA, the Frobenius norm is employed to calculate the reconstruction error when projecting the data onto a lower-dimensional subspace. The projection matrix used in PCA and all similar dimensionality reduction techniques is assumed to be orthonormal, or equivalently, it lies on the Stiefel manifold. The Stiefel manifold is an example of a smooth manifold. As noted by Lee (2012), a smooth manifold is a space that "locally looks like" $\mathbb{R}^n$ and we can perform calculus on it.

**Definition 2** (Stiefel Manifold). *The Stiefel manifold, herein denoted as $\mathcal{S}(p, q)$, is defined as the space consisting of all $p \times q$ orthonormal matrices,*

$$\mathcal{S}(p, q) = \{W \in \mathbb{R}^{p \times q} : W^\top W = I_q\}.$$

On a differentiable manifold, such as the Stiefel manifold, an important notion is the tangent space.

**Definition 3** (Tangent Space). *Let $\mathcal{M}$ be a smooth manifold and let $p \in \mathcal{M}$. The tangent space at $p$, denoted as $\mathcal{T}_p \mathcal{M}$, is the set of all possible derivatives of smooth curves passing through $p$. For the Stiefel manifold, this can be defined as*

$$\mathcal{T}_W \mathcal{S}(p, q) = \{V \in \mathbb{R}^{p \times q} : V^\top W + W^\top V = 0\}.$$

A recurrent term in manifold theory is that of the Riemannian gradient. First, recall that the Euclidean gradient, or standard gradient, is the vector consisting of the partial derivatives of a real function $f \colon \mathbb{R}^n \to \mathbb{R}$ with respect to each coordinate direction. The Riemannian gradient generalises this concept to manifolds.

**Definition 4** (Riemannian Gradient). *Consider a smooth function $f \colon \mathcal{M} \to \mathbb{R}$, where $\mathcal{M}$ is a smooth manifold. The Riemannian gradient, denoted $\mathrm{grad}_f$, is defined as the unique tangent vector in $\mathcal{T}_x \mathcal{M}$ satisfying*

$$\langle \mathrm{grad}_f(x), v \rangle_x = Df(x)[v], \quad \forall v \in \mathcal{T}_x \mathcal{M},$$

*where $Df(x)[v]$ is the directional derivative of $f$ at $x$ in the direction $v$.*

Throughout, we equip the Stiefel manifold $\mathcal{S}(p, q)$ with the *embedded* (Euclidean) metric inherited from $\mathbb{R}^{p \times q}$ (Absil et al., 2008),

$$\langle Z_1, Z_2 \rangle = \mathrm{tr}(Z_1^\top Z_2), \quad Z_1, Z_2 \in \mathcal{T}_W \mathcal{S}(p, q). \tag{5}$$

Under this metric, the Riemannian gradient of a smooth function $f \colon \mathcal{S}(p, q) \to \mathbb{R}$ is obtained by projecting the Euclidean gradient $\nabla f(W)$ onto the tangent space $\mathcal{T}_W \mathcal{S}(p, q)$,

$$\mathrm{grad}\, f(W) = \mathrm{Proj}_W\big(\nabla f(W)\big) = \nabla f(W) - W \,\mathrm{sym}\big(W^\top \nabla f(W)\big), \tag{6}$$

where $\text{sym}(A) = (A + A^\top)/2$ and $\text{Proj}_W$ denotes the orthogonal projection onto the tangent space. The residual $W \text{sym}(W^\top \nabla f(W))$ lies in the *normal space* $(\mathcal{T}_W \mathcal{S}(p, q))^\perp = \{WS : S^\top = S\}$, which is orthogonal to every tangent vector under (5). Consequently, for any $V \in \mathcal{T}_W \mathcal{S}(p, q)$,

$$\langle \text{grad} f(W),\, V \rangle = \langle \nabla f(W),\, V \rangle, \tag{7}$$

so whenever the second argument is tangent, the Euclidean gradient, $\nabla f(W)$, may replace the Riemannian gradient, $\text{grad} f(W)$, inside the inner product. This property will be used in Section 4.3 when formulating the optimisation subproblem.

When solving optimisation problems constrained on a manifold it is crucial to ensure that each iterative step remains on the manifold regardless of the operation taking place. Retraction ensures that tangent vectors are "retracted" back to the manifold, ensuring feasibility of iterates.

**Definition 5** (Retraction (Absil et al., 2008)). *A retraction on a smooth manifold $\mathcal{M}$ is a smooth mapping that assigns to each point $W \in \mathcal{M}$ a map $\text{Retr}_W : \mathcal{T}_W \mathcal{M} \to \mathcal{M}$ satisfying:*

*(i) $\text{Retr}_W(0_W) = W$, where $0_W$ denotes the zero element of $\mathcal{T}_W \mathcal{M}$.*

*(ii) The differential of $\text{Retr}_W$ at $0_W$ satisfies $D \text{Retr}_W(0_W) = \text{id}_{\mathcal{T}_W \mathcal{M}}$,*

*where $\text{id}_{\mathcal{T}_W \mathcal{M}}$ denotes the identity mapping on $\mathcal{T}_W \mathcal{M}$.*

Condition (i) ensures that the current point is a fixed point of the retraction, while condition (ii) ensures that the retraction is locally identical to the identity map to first order; equivalently, for any $V \in \mathcal{T}_W \mathcal{M}$, the curve $t \mapsto \text{Retr}_W(tV)$ has initial velocity $V$. For the Stiefel manifold, a popular retraction choice, which we adopt in our formulation, is the polar retraction,

$$\text{Retr}_W(V) = (W + V)(I_q + V^\top V)^{-1/2}. \tag{8}$$

Sparse methods that make use of the $\ell_1$ norm, e.g. LASSO regression (Tibshirani, 1996), face the issue of non-smoothness since the $\ell_1$ norm is non-differentiable. To overcome this issue we make use of proximal mappings, which are well-defined under mild assumptions.

**Definition 6** (Proximal Mapping). *Let $f$ denote a closed, convex function. The proximal mapping associated to $f$ is defined as*

$$\text{prox}_f(Y) := \arg\min_X f(X) + \frac{1}{2}\|X - Y\|_{\mathcal{F}}^2.$$

**Theorem 1** (Proximal Mapping Existence–Uniqueness (Parikh & Boyd, 2014)). *Let $f : \mathbb{R}^{n \times m} \to \mathbb{R} \cup \{+\infty\}$ be a closed, convex, and lower semicontinuous function. Then, for every $Y \in \mathbb{R}^{n \times m}$, the proximal mapping*

$$\text{prox}_f(Y) := \arg\min_X\ f(X) + \frac{1}{2}\|X - Y\|_{\mathcal{F}}^2$$

*has a unique minimiser; that is, $\text{prox}_f$ is well-defined and single-valued on $\mathbb{R}^{n \times m}$.*

The proof for this Theorem is provided in the Appendix. The objective function we propose belongs in the category of convex but non-differentiable functions. As a result, the application of gradient descent becomes infeasible. Instead, we consider proximal gradient descent, specifically over the Stiefel manifold, that seeks to optimise a non-differentiable but decomposable function. The update rule for proximal gradient descent in the Euclidean setting (Combettes & Wajs, 2005) is provided in the following definition.

**Definition 7** (Proximal Gradient Step). *Assume $f$ is a decomposable function, $f(X) = g(X) + h(X)$, where $g$ is convex and differentiable and $h$ is convex, non-differentiable. The proximal gradient update rule is defined as*

$$X_{k+1} := \arg\min_Z g(X_k) + \langle \nabla g(X_k), Z - X_k \rangle + \frac{1}{2t}\|Z - X_k\|_F^2 + h(Z), \tag{9}$$

*where $t > 0$ is a stepsize parameter.*

The update rule of the proximal gradient descent can be equivalently written in terms of the proximal mapping as

$$X_{k+1} = \text{prox}_{th}\left(X_k - t\nabla g(X_k)\right).$$

# 4 Sparse Covariance Supervised PCA (SCS-PCA)

In this section, we present our proposal for sparse supervised PCA which we call sparse covariance supervised PCA or sparse CSPCA (SCS-PCA). First, we discuss the motivation behind our method before proceeding with the mathematical formulation of SCS-PCA. Next, we present the optimisation algorithm we employ to derive the projection matrix with sparse loadings and discuss the convergence of it. The section concludes with some practical considerations.

## 4.1 Motivation

The method we propose herein seeks to provide a direct trade-off between predictive accuracy and sparsity performance in a single objective function via a tunable hyperparameter and it is accompanied by a robust optimisation scheme that provides theoretical guarantees for global convergence to a stationary point. The main innovation of our proposal lies in the direct solution of the objective function without relying on auxiliary objective functions, through the use of manifold optimisation that confronts the non-convex optimisation problem directly, without resorting to biconvex formulations.

Specifically, we apply an $\ell_1$ penalty on the loadings of the projection matrix seeking to trade-off CSPCA's predictive performance and sparsity induced by the lasso penalty. To achieve this, we introduce a tunable parameter that controls the magnitude of sparsity imposed on CSPCA's objective function and hence controls the shrinkage of the loadings. Following CSPCA, our method can be applied for any type of response variable. Unlike SSPCA, which requires the specification of a kernel family and associated bandwidth parameters for the response (e.g. the radial basis function kernel), SCS-PCA incorporates the response through the data-response covariance $X^\top YY^\top X$ directly, avoiding kernel selection. The balancing parameter $\kappa$ in $C = X^\top YY^\top X + \kappa X^\top X$ controls the trade-off between supervised and unsupervised objectives and is tuned via cross-validation at the initialisation step (see Section 4.6), which adds negligible cost since it requires a single eigendecomposition of $C$ for each candidate value. The derived optimisation problem is non-convex and non-smooth. We employ manifold proximal gradient descent (ManPG) (Chen et al., 2020) which can handle non-convexity and non-smoothness simultaneously by decomposing the objective function into a smooth and non-smooth part. The ManPG algorithm is an extension of traditional proximal gradient descent onto the Stiefel manifold, where the projection matrix of the proposed optimisation problem lies. We provide theoretical guarantees for the global convergence of the algorithm to a stationary point.

## 4.2 Mathematical Formulation

Our proposal is based on CSPCA's formulation presented in (2). We introduce a slightly different notation to the optimisation problem by specifying explicitly that the projection matrix subsides in the Stiefel manifold,

$$\max_{W \in \mathcal{S}(p,q)} \text{tr}\left(W^\top CW\right),$$

where $C = X^\top YY^\top X + \kappa X^\top X$ defined as before. For classification tasks, a delta kernel, $\delta(y, y')$ is used to model the response instead of $YY^\top$ in the objective function. To induce sparsity, we apply an $\ell_1$ penalty (Tibshirani, 1996) directly on the loadings of the projection matrix, which allows for a direct trade-off between prediction accuracy and sparsity. This way we expect a large number of non-significant features is eliminated from the projected data, through shrinkage, enhancing interpretability. The sparse optimisation problem can thus be defined as

$$\max_{W \in \mathcal{S}(p,q)} \text{tr}\left(W^\top CW\right) - \eta\|W\|_1, \tag{10}$$

where $\eta > 0$ is the parameter controlling the magnitude of sparsity. The proposed method can be applied for any type of response variable and applies sparsity directly on the loadings in the form of an $\ell_1$ penalty

rather than a constrained optimisation problem. Unlike existing methods, we seek to optimise (10) directly without using surrogate reformulations. Specifically, we employ the ManPG algorithm, which combines manifold optimisation and proximal gradient descent theory to simultaneously handle the orthogonality and sparsity constraints and prove it leads to global convergence on a stationary point.

### 4.3  Optimisation

Since the objective function in (10) is non-differentiable, proximal gradient descent algorithms are required to derive the projection matrix with sparse loadings. Hence we need to decompose the objective into a smooth, potentially convex part and a non-smooth but convex part to define a problem of the form,

$$\min F(W) := f(W) + h(W),$$

where $f$ is smooth and $h$ is convex, non-smooth. By rewriting (10) as a minimisation problem,

$$\min_{W \in \mathcal{S}(p,q)} F(W) := -\text{tr}(W^\top C W) + \eta \|W\|_1,$$

we can observe that $f(W) = -\text{tr}(W^\top C W)$ is indeed smooth, while $h(W) = \eta \|W\|_1$ is convex and non-smooth, since $h$ is non-differentiable at 0. The Euclidean gradient of the smooth part can easily be calculated as

$$\nabla f(W) = -2CW = -2(X^\top Y Y^\top X + \kappa X^\top X)W.$$

Using the proximal gradient update rule from Definition 7 on (10), we derive

$$W_{k+1} = \arg\min_Z -\text{tr}(W_k^\top C W_k) + \langle -2CW_k, Z - W_k \rangle + \frac{1}{2t}\|Z - W_k\|_{\mathcal{F}}^2 + \eta\|Z\|_1, \tag{11}$$

where $t > 0$ is a stepsize parameter. Since $W$ lies on the Stiefel manifold, $\mathcal{M} = \mathcal{S}(p,q)$, we need to ensure that the descent direction lies in the tangent space, $\mathcal{T}_W \mathcal{M}$, hence we need to employ a manifold proximal gradient descent algorithm instead. Chen et al. (2020) proposed the following subproblem to transition from the Euclidean setting to the Stiefel manifold,

$$V_k := \arg\min_V \langle \text{grad}\, f(W_k), V \rangle + \frac{1}{2t}\|V\|_{\mathcal{F}}^2 + \eta\|W_k + V\|_1, \tag{12}$$

such that $V \in \mathcal{T}_{W_k}\mathcal{M}$, where $\text{grad}\, f$ is the Riemannian gradient. By (7), we have

$$\langle \text{grad}\, f(W_k), V \rangle = \langle \nabla f(W_k), V \rangle = \langle -2CW_k, V \rangle, \quad \forall\, V \in \mathcal{T}_{W_k}\mathcal{M},$$

which allows us to rewrite the subproblem in (12) as

$$V_k := \arg\min_V \langle -2CW_k, V \rangle + \frac{1}{2t}\|V\|_{\mathcal{F}}^2 + \eta\|W_k + V\|_1, \tag{13}$$

such that $V \in \mathcal{T}_{W_k}\mathcal{M}$. As a result, there is no need to calculate the Riemannian gradient, but the Euclidean. The corresponding projection matrix at step $k+1$ can be derived by retraction,

$$W_{k+1} = \text{Retr}(\alpha V_k),$$

where $\alpha$ is determined by an Armijo line search. Popular choices of retraction for the Stiefel manifold include the exponential mapping, the polar decomposition, the QR decomposition and the Cayley transformation. In our implementation, we employ the polar decomposition defined in (8).

### 4.4  Solving SCS-PCA using RSSN

To solve a subproblem of the form as in (13), Chen et al. (2020) suggested the use of the Regularised Semi-Smooth Newton (RSSN) method, which, applied to SCS-PCA, is guaranteed to converge. First, we denote $A_k(V) = V^\top W_k + W_k^\top V$, and then rewrite the subproblem in (12) as

$$V_k := \arg\min_V \langle -2CW_k, V \rangle + \frac{1}{2t}\|V\|_{\mathcal{F}}^2 + \eta\|W_k + V\|_1$$

such that $A_k(V) = 0$. Let $\mathcal{L}(V; \Lambda)$ denote the Langrangian function of the subproblem's objective function,

$$\mathcal{L}(V; \Lambda) = \langle -2CW_k, V \rangle + \frac{1}{2t}\|V\|_{\mathcal{F}}^2 + \eta\|W_k + V\|_1 - \langle A_k(V), \Lambda \rangle.$$

To solve the subproblem, we define the Karush-Kuhn-Tucker (KKT) conditions,

$$0 \in \partial_V \mathcal{L}(V; \Lambda) \quad \text{and} \quad A_k(V) = 0.$$

The proximal mapping point of $h = \eta\|\cdot\|_1$ at point $W_k$ is defined as

$$\text{prox}_{t\eta\|\cdot\|_1}(W_k) = \arg\min_Z \frac{1}{2}\|Z - W_k\|_F^2 + t\eta\|Z\|_1. \tag{14}$$

Using the proximal mapping we can write

$$V(\Lambda) = \text{prox}_{th}(B(\Lambda)) - W_k,$$

where $B(\Lambda) = W_k + t(2CW_k - A_k^*(\Lambda))$, and $A_k^*(\Lambda)$ is the adjoint operator of $A_k$.

In the case of the $\ell_1$ norm, the proximal mapping is defined as the soft-thresholding operator, hence there is no need to solve (14). The soft-thresholding operator is defined as

$$\text{prox}_{t\eta\|\cdot\|_1}(Y)_{ij} = \text{sign}(Y_{ij}) \max(|Y_{ij}| - t\eta, 0).$$

We use the RSSN method to solve $\mathcal{E}(\Lambda) = A_k(V(\Lambda)) = 0$ to derive the $k$-th iterate of the subproblem, $V_k$. Specifically, we compute the Newton direction $d$ by solving

$$(G(\Lambda) + \rho I)d = -\text{vec}(\mathcal{E}(\Lambda)),$$

where $G(\Lambda)$ is a representation of the generalised Jacobian of $\mathcal{E}$ and $\rho > 0$ is a regularisation parameter. The updates for $\Lambda$ are calculated using the following rule,

$$\Lambda_{k+1} \leftarrow \Lambda_k + d.$$

Thus, we can easily solve the subproblem and derive $V_k$. We can then update $W_k$ using a retraction step, i.e. $W_{k+1} = \text{Retr}(\alpha V_k)$, where $\alpha$ is determined by an Armijo line search. An overview of the steps to solve SCS-PCA is given in Algorithm 1.

---

**Algorithm 1** SCS-PCA using ManPG Algorithm

---

**Require:** Initial $W_0 \in \mathcal{M}$ (CSPCA), $\gamma \in (0, 1)$, stepsize $t > 0$
**Ensure:** Projection matrix $W$.
1: **for** $k = 0, 1, \ldots$ **do**
2:     Obtain $V_k$ by solving the subproblem,
       $V_k := \arg\min_{V \in \mathcal{T}_{W_k}\mathcal{M}} \langle -2CW_k, V \rangle + \frac{1}{2t}\|V\|_{\mathcal{F}}^2 + \eta\|W_k + V\|_1$, where $C = X^\top YY^\top X + \kappa X^\top X$.
3:     Set $\alpha = 1$
4:     **while** $F(\text{Retr}_{W_k}(\alpha V_k)) > F(W_k) - \frac{\alpha\|V_k\|_F^2}{2t}$ **do**
5:         $\alpha \leftarrow \gamma\alpha$
6:     **end while**
7:     Update the iterate using retraction:
       $W_{k+1} = \text{Retr}_{W_k}(\alpha V_k)$.
8: **end for**

---

### 4.5 Convergence of SCS-PCA and practical considerations

The convergence of Algorithm 1 is guaranteed by the ManPG framework of Chen et al. (2020) which establishes that every accumulation point of the iterates is a stationary point of (10), provided the smooth term has Lipschitz continuous gradient, the non-smooth term is convex, and the retraction satisfies standard boundedness conditions. We now formalise this for SCS-PCA.

**Theorem 2** (SCS-PCA Convergence). *Consider the optimisation problem,*

$$\min_{W \in \mathcal{S}(p,q)} F(W) := \underbrace{-\operatorname{tr}(W^\top C W)}_{f(W)} + \underbrace{\eta \|W\|_1}_{h(W)},$$

*where $C = X^\top Y Y^\top X + \kappa X^\top X$ is symmetric positive semi-definite. Suppose the following conditions hold:*

*(i) $f$ is smooth on $\mathbb{R}^{p \times q}$ and its Euclidean gradient $\nabla f(W) = -2CW$ is Lipschitz continuous with constant $L = 2\|C\|_{\text{op}} < \infty$;*

*(ii) $h$ is convex and Lipschitz continuous;*

*(iii) the retraction $\operatorname{Retr}_W$ (Definition 5) satisfies, for all $W \in \mathcal{S}(p,q)$ and $V \in \mathcal{T}_W \mathcal{S}(p,q)$,*

$$\|\operatorname{Retr}_W(V) - W\|_{\mathcal{F}} \leq M_1 \|V\|_{\mathcal{F}},$$
$$\|\operatorname{Retr}_W(V) - (W + V)\|_{\mathcal{F}} \leq M_2 \|V\|_{\mathcal{F}}^2,$$

*for finite constants $M_1, M_2 > 0$;*

*(iv) the step size satisfies $0 < t \leq 1/L$ and the line-search parameter $\gamma \in (0,1)$.*

*Then the sequence $\{W_k\}$ generated by Algorithm 1 converges to a stationary point of $F$, in the sense that every accumulation point $W^*$ of $\{W_k\}$ satisfies*

$$0 \in \operatorname{grad} f(W^*) + \operatorname{Proj}_{\mathcal{T}_{W^*}\mathcal{S}} \, \partial h(W^*).$$

The detailed proof is provided in the Appendix.

In practice, Algorithm 1 can be initialised from any point on the Stiefel manifold and Theorem 2 guarantees convergence to a stationary point regardless of the choice. However, since the problem is non-convex, different initialisations may lead to different stationary points. We recommend initialising with the solution to the standard CSCPA, i.e., the top $q$ eigenvectors of $C$, as this provides a warm start near a good region of the objective landscape at the cost of a single eigendecomposition. Empirically, we observe that random Stiefel initialisations lead to stationary points with higher prediction error on average, confirming the practical benefit of the CSPCA warm start (see Appendix A.2). Overall, the per-iteration cost is approximately $O(p^2 q)$, dominated by the gradient computation. The RSSN method for solving the subproblem performs robustly and efficiently (Chen et al., 2020), and its cost remains negligible relative to the gradient computation in the regime $q \ll p$ considered throughout this work.

### 4.6 Tuning the sparsity parameter

The proposed formulation in (10) consists of two parameters, the balancing parameter, $\kappa$, and the sparsity inducing parameter $\eta$. We observe empirically that CSPCA is not very sensitive to the choice of $\kappa$; large fluctuations in its value are needed before a significant change in performance is observed. We therefore suggest tuning $\kappa$ at the initialisation step, where $W_0$ is taken as the CSPCA solution for each candidate $\kappa$ via eigendecomposition of $C$, and selecting the value that minimises the validation error. Since only an eigendecomposition is required (and not the full ManPG algorithm), this step is computationally inexpensive. The selected $\kappa$ is then held fixed when tuning the sparsity parameter $\eta$ via $K$-fold cross-validation as described in Algorithm 2.

## 5 Experimental Results

### 5.1 Simulations

We conduct a series of simulation analyses to examine the performance of SCS-PCA against existing methods in sparse supervised PCA, supervised PCA as well as baseline dimensionality reduction methods such as

---

**Algorithm 2** K-fold Cross-Validation for Tuning the Sparsity Parameter $\eta$

---

**Require:** Data $(X, Y)$, candidate values $\{\eta_1, \ldots, \eta_L\}$, number of folds $K$
**Ensure:** Optimal sparsity parameter $\eta^*$
  1: Partition data into $K$ equal folds
  2: **for** $i = 1, \ldots, L$ **do**                                                 ▷ Loop over grid of $\eta$ values
  3:     Set $\eta = \eta_i$
  4:     Initialise validation error list: $E_i = [\cdot]$
  5:     **for** $k = 1, \ldots, K$ **do**                                   ▷ K-fold cross-validation
  6:         Use fold $k$ as validation set; remaining $K{-}1$ folds as training set
  7:         Train Algorithm 1 on training data with current $\eta$
  8:         Compute prediction $\hat{Y}_{\text{val}}$ on validation set
  9:         Compute MSE: $e_k = \frac{1}{n_k}\|Y_{\text{val}} - \hat{Y}_{\text{val}}\|_{\mathcal{F}}^2$
10:         Append $e_k$ to $E_i$
11:     **end for**
12:     Compute mean validation error: $\bar{E}_i = \frac{1}{K}\sum_{k=1}^{K} E_i[k]$
13: **end for**
14: Select $\eta^* = \arg\min_{\eta_i} \bar{E}_i$

---

PCA and PLS. Specifically, for supervised PCA, we consider Bair's method (Bair et al., 2006), supervised PCA using HSIC (Barshan et al., 2011) and CSPCA (Papazoglou & Yin, 2025), while for sparse supervised PCA we consider, sparse PLS (SPLS) (Chun & Keleş, 2010), sparse PCA (SPCA) (Zou et al., 2006) and sparse supervised PCA (SSPCA) (Sharifzadeh et al., 2017). We evaluate prediction accuracy (MSE), sparsity (number of selected variables), variable selection quality (precision, recall, F1, TP, FP, FN relative to the true support), and statistical significance of pairwise MSE differences via paired Wilcoxon signed-rank tests across dataset replicates. Additionally, we examine the full sparsity–prediction trade-off via Pareto front analysis for all sparse supervised methods (SSPCA, SPLS, SCS-PCA, Appendix A.3).

We consider two different simulation models, a linear and a non-linear model. In each setting, we generate 100 independent datasets with $n = 100$ observations and $p = 500$ features, where we assume each true underlying model depends only on the first four features, creating a high-dimensional sparse setting. The specific models are defined as

- Simulation 1: $Y = 3X_1 - 2X_2 - 5X_3 + 4X_4 + \epsilon$,

- Simulation 2: $Y = e^{X_1} + 4\sin(X_2) - 3X_3 + X_4 + \epsilon$,

where $\epsilon \sim N(0, 0.1^2)$ corresponds to random noise and the true support is $\{X_1, X_2, X_3, X_4\}$ in all three settings. For the data generating mechanism of both simulations we consider two scenarios. First, the data are independently and identically distributed (i.i.d.) from a standard normal distribution, i.e. $X \sim N(0, I_p)$. Second, we introduce correlation between features to align closer with realistic scenarios, e.g. microarray data. Specifically, we generate the data from a normal distribution with mean zero and covariance matrix $\Sigma$, defined as a Toeplitz matrix, that is $\Sigma = \Sigma_{ij} = \rho^{|i-j|}$. We set $\rho = 0.7$, inducing strong correlation between features, such that the correlation between $X_i$ and $X_j$ decays exponentially with $|i-j|$. All simulations were performed for $q = 2, 3$ and 4 components and results are averaged across the 100 dataset replicates using Monte Carlo estimates along with the respective standard errors. For each dataset a $60\% - 20\% - 20\%$ split into training, validation and test sets was performed. To ensure a fair comparison, all methods share the same training–validation–test split and standardisation. Hyperparameters are selected by minimising validation MSE over method-specific grids detailed in the supplementary material. All projection-based methods (PCA, HSIC, Bair's, CSPCA, SCS-PCA, SPCA, SSPCA) follow the same prediction pipeline, where the data are projected via $Z = XW$, ordinary least squares are fitted on the projected training data and predictions are made on the test set. For SPLS, predictions are obtained directly via $\hat{Y} = X\hat{\beta}$.

Table 1 presents the results for the first simulation for the i.i.d. (top) and correlated (bottom) scenarios. SCS-PCA achieves the strongest sparsity across all settings, selecting on average 4.6 variables in the i.i.d. scenario and 3.0 in the correlated scenario — closest to the true support size of 4 — while maintaining the

Table 1: Simulation 1 results across 100 independent datasets for the I.I.D. (top) and correlated (bottom) scenarios. MSE with standard errors, number of selected variables (Select), precision (Prec.) and recall (Rec.) are reported as Monte Carlo averages.

| Method | $q = 2$ | | | | $q = 3$ | | | | $q = 4$ | | | |
|---|---|---|---|---|---|---|---|---|---|---|---|---|
| | MSE (s.e.) | Select | Prec. | Rec. | MSE (s.e.) | Select | Prec. | Rec. | MSE (s.e.) | Select | Prec. | Rec. |
| *I.I.D. Scenario* | | | | | | | | | | | | |
| PCA† | 1.010 (0.036) | 500.0 | — | — | 1.006 (0.036) | 500.0 | — | — | 1.008 (0.036) | 500.0 | — | — |
| PLS† | 0.918 (0.033) | 500.0 | — | — | 0.917 (0.033) | 500.0 | — | — | 0.918 (0.033) | 500.0 | — | — |
| HSIC† | 1.009 (0.037) | 500.0 | — | — | 1.000 (0.037) | 500.0 | — | — | 0.995 (0.036) | 500.0 | — | — |
| Bair† | 0.959 (0.034) | 225.3 | — | — | 0.960 (0.035) | 225.3 | — | — | 0.957 (0.034) | 225.3 | — | — |
| CSPCA† | 0.937 (0.034) | 500.0 | — | — | 0.938 (0.034) | 500.0 | — | — | 0.941 (0.034) | 500.0 | — | — |
| SPLS‡ | **0.121** (0.028) | 7.7 | **0.78** | **0.96** | **0.152** (0.030) | 22.4 | 0.42 | **1.00** | **0.224** (0.033) | 43.5 | 0.18 | **1.00** |
| SPCA† | 1.025 (0.037) | 66.7 | 0.01 | 0.15 | 1.034 (0.038) | 97.0 | 0.01 | 0.23 | 1.029 (0.038) | 126.0 | 0.01 | 0.25 |
| SSPCA† | 1.021 (0.040) | 80.5 | 0.02 | 0.28 | 1.005 (0.043) | 85.9 | 0.02 | 0.32 | 1.013 (0.042) | 92.8 | 0.04 | 0.32 |
| *SCS-PCA* | 0.405 (0.026) | **4.6** | 0.57 | 0.49 | 0.408 (0.027) | **5.8** | **0.46** | 0.49 | 0.415 (0.028) | **6.6** | **0.34** | 0.49 |
| *Correlated Scenario ($\rho = 0.7$)* | | | | | | | | | | | | |
| PCA† | 1.059 (0.055) | 500.0 | — | — | 1.058 (0.054) | 500.0 | — | — | 1.060 (0.055) | 500.0 | — | — |
| PLS† | 1.139 (0.053) | 500.0 | — | — | 1.145 (0.052) | 500.0 | — | — | 1.148 (0.052) | 500.0 | — | — |
| HSIC† | 1.067 (0.055) | 500.0 | — | — | 1.075 (0.055) | 500.0 | — | — | 1.070 (0.053) | 500.0 | — | — |
| Bair† | 1.082 (0.052) | 224.2 | — | — | 1.080 (0.052) | 224.2 | — | — | 1.082 (0.052) | 224.2 | — | — |
| CSPCA† | 1.066 (0.051) | 500.0 | — | — | 1.069 (0.052) | 500.0 | — | — | 1.067 (0.052) | 500.0 | — | — |
| SPLS‡ | **0.447** (0.038) | 8.5 | **0.78** | **0.69** | **0.526** (0.042) | 33.6 | 0.41 | **0.76** | **0.555** (0.037) | 33.3 | 0.31 | **0.79** |
| SPCA† | 1.056 (0.056) | 52.7 | 0.01 | 0.09 | 1.073 (0.055) | 76.5 | 0.01 | 0.11 | 1.085 (0.057) | 100.7 | 0.01 | 0.21 |
| SSPCA† | 1.058 (0.057) | 98.7 | 0.03 | 0.32 | 1.082 (0.060) | 83.2 | 0.02 | 0.29 | 1.083 (0.063) | 77.7 | 0.03 | 0.28 |
| *SCS-PCA* | 0.613 (0.033) | **3.0** | 0.67 | 0.36 | 0.612 (0.033) | **3.9** | **0.52** | 0.38 | 0.619 (0.034) | **4.8** | **0.44** | 0.39 |

† SCS-PCA significantly better ($p < 0.001$, paired Wilcoxon test). ‡ SPLS significantly better than SCS-PCA across all $q$ ($p < 0.01$).

highest precision among all methods (0.57 and 0.67, respectively, at $q = 2$). In terms of prediction, SCS-PCA significantly outperforms all projection-based and supervised methods, including SPCA and SSPCA, across all component counts and both scenarios ($p < 0.001$). SPLS achieves the lowest MSE overall, which is expected given that it is a regression-based method operating directly in the original feature space. Nevertheless, SPLS selects substantially more variables than SCS-PCA, particularly as $q$ increases (e.g. 43.5 vs 6.6 at $q = 4$ in the i.i.d. case), with a corresponding decline in precision, where SCS-PCA outperforms SPLS for $q = 3$ and 4 components for both i.i.d. and correlated scenarios.

Numerical results for the i.i.d. and correlated scenarios of Simulation 2 are presented in Table 1. SCS-PCA again achieves the strongest sparsity (6.6 variables in i.i.d., 3.0 in the correlated scenario at $q = 2$) and the highest precision among all methods in all but one setting ($q = 2$). Notably, while SPLS retains the best MSE, its precision degrades sharply as the number of components increases — from 0.75 to 0.19 in the i.i.d. case — reflecting a substantial increase in false positives (from 9.2 to 49.3 selected variables). SCS-PCA, by contrast, remains stable across all $q$, with selected variables staying close to the true support size. Among projection-based and supervised methods, SCS-PCA significantly outperforms all methods in both MSE and variable selection quality across all settings ($p < 0.001$) for both i.i.d. and correlated scenarios.

Across both simulations and both data-generating scenarios, a consistent pattern emerges. SCS-PCA produces the sparsest solutions among all methods, selecting a number of variables closest to the true support size, and achieves the highest precision in nearly every setting. In terms of prediction, SCS-PCA significantly outperforms all projection-based and supervised PCA methods — including SPCA, SSPCA, CSPCA and HSIC — across all component counts ($p < 0.001$). The only method achieving lower MSE is SPLS, a regression-based approach that operates directly in the original feature space rather than through a learned projection. However, SPLS consistently selects substantially more variables than SCS-PCA, particularly for $q \geq 3$, with a corresponding decline in precision. These results illustrate the trade-off offered by SCS-PCA's objective function: competitive prediction accuracy with markedly more interpretable projections.

Table 2: Simulation 2 results across 100 independent datasets for the I.I.D. (top) and correlated (bottom) scenarios. MSE with standard errors, number of selected variables (Select), precision (Prec.) and recall (Rec.) are reported as Monte Carlo averages.

| Method | $q = 2$ MSE (s.e.) | Select | Prec. | Rec. | $q = 3$ MSE (s.e.) | Select | Prec. | Rec. | $q = 4$ MSE (s.e.) | Select | Prec. | Rec. |
|---|---|---|---|---|---|---|---|---|---|---|---|---|
| *I.I.D. Scenario* | | | | | | | | | | | | |
| PCA[†] | 1.015 (0.045) | 500.0 | — | — | 1.013 (0.045) | 500.0 | — | — | 1.015 (0.044) | 500.0 | — | — |
| PLS[†] | 0.948 (0.041) | 500.0 | — | — | 0.949 (0.041) | 500.0 | — | — | 0.949 (0.041) | 500.0 | — | — |
| HSIC[†] | 1.002 (0.044) | 500.0 | — | — | 0.992 (0.043) | 500.0 | — | — | 0.989 (0.044) | 500.0 | — | — |
| Bair[†] | 0.973 (0.042) | 226.1 | — | — | 0.975 (0.042) | 226.1 | — | — | 0.976 (0.042) | 226.1 | — | — |
| CSPCA[†] | 0.963 (0.043) | 500.0 | — | — | 0.962 (0.042) | 500.0 | — | — | 0.963 (0.042) | 500.0 | — | — |
| SPLS[‡] | **0.245** (0.023) | 9.2 | **0.75** | **0.86** | **0.270** (0.0262) | 23.6 | 0.38 | **0.88** | **0.350** (0.031) | 49.3 | 0.19 | **0.98** |
| SPCA[†] | 1.032 (0.044) | 65.4 | 0.01 | 0.15 | 1.024 (0.045) | 97.8 | 0.01 | 0.23 | 1.037 (0.045) | 126.2 | 0.01 | 0.27 |
| SSPCA[†] | 0.993 (0.048) | 63.3 | 0.06 | 0.24 | 0.998 (0.048) | 88.3 | 0.02 | 0.28 | 1.028 (0.050) | 71.6 | 0.03 | 0.24 |
| *SCS-PCA* | 0.503 (0.033) | **6.6** | 0.52 | 0.46 | **0.511** (0.034) | **7.7** | **0.42** | 0.46 | 0.514 (0.034) | **8.7** | **0.30** | 0.42 |
| *Correlated Scenario ($\rho = 0.7$)* | | | | | | | | | | | | |
| PCA[†] | 1.059 (0.055) | 500.0 | — | — | 1.058 (0.054) | 500.0 | — | — | 1.060 (0.055) | 500.0 | — | — |
| PLS[†] | 1.139 (0.053) | 500.0 | — | — | 1.145 (0.052) | 500.0 | — | — | 1.148 (0.052) | 500.0 | — | — |
| HSIC[†] | 1.067 (0.055) | 500.0 | — | — | 1.075 (0.055) | 500.0 | — | — | 1.070 (0.053) | 500.0 | — | — |
| Bair[†] | 1.082 (0.052) | 224.2 | — | — | 1.080 (0.052) | 224.2 | — | — | 1.082 (0.052) | 224.2 | — | — |
| CSPCA[†] | 1.066 (0.051) | 500.0 | — | — | 1.069 (0.052) | 500.0 | — | — | 1.067 (0.052) | 500.0 | — | — |
| SPLS[‡] | **0.447** (0.038) | 8.5 | **0.78** | **0.69** | **0.526** (0.042) | 33.6 | 0.41 | **0.76** | **0.555** (0.037) | 33.3 | 0.31 | **0.79** |
| SPCA[†] | 1.056 (0.056) | 52.7 | 0.01 | 0.09 | 1.073 (0.055) | 76.5 | 0.01 | 0.11 | 1.085 (0.057) | 100.7 | 0.01 | 0.21 |
| SSPCA[†] | 1.058 (0.057) | 98.7 | 0.03 | 0.32 | 1.082 (0.060) | 83.2 | 0.02 | 0.29 | 1.083 (0.063) | 77.7 | 0.03 | 0.28 |
| *SCS-PCA* | 0.613 (0.033) | **3.0** | 0.67 | 0.36 | 0.612 (0.033) | **3.9** | **0.52** | 0.38 | 0.619 (0.034) | **4.8** | **0.44** | 0.39 |

[†] SCS-PCA significantly better ($p < 0.001$, paired Wilcoxon test). [‡] SPLS significantly better than SCS-PCA across all $q$ ($p < 0.01$).

Table 3: Numerical Results from the Mice Toxicity Dataset for $q = 2$ components.

| Method | MSE (s.e.) | Non-zero variables (s.e.) |
|---|---|---|
| PCA | 1.0681 (0.1701) | 3116.0 (0.0) |
| PLS | 0.6194 (0.0725) | 3116.0 (0.0) |
| HSIC | 0.9878 (0.1502) | 3115.1 (0.2) |
| Bair | 0.9545 (0.1650) | 2123.9 (34.3) |
| CSPCA | 0.7249 (0.1164) | 3115.1 (0.2) |
| SPCA | 1.1087 (0.1766) | 11.4 (0.9) |
| SPLS | 0.6978 (0.1042) | 982.4 (355.1) |
| SSPCA | 0.9285 (0.1367) | 417.6 (152.8) |
| **SCS-PCA** | 0.6009 (0.0835) | 246.1 (34.8) |

## 5.2 Applications on Real Datasets

We use three real world, high-dimensional genetic datasets, one with a continuous phenotype and two with a binary phenotype, to examine SCS-PCA's performance against existing methods discussed in Section 2. For regression tasks, we employ the mice liver toxicity genetic dataset from Bushel et al. (2007), which is publicly available from R's Bioconductor package. The same methods used during the simulations in the previous section were also considered for this analysis. For classification, we employ the well-known leukemia dataset by Golub et al. (1999) — publicly available in Kaggle — which seeks to classify patients into acute myeloid leukemia (AML) and acute lymphoblastic leukemia (ALL), along with a colon cancer dataset extracted from the colonCA library from the Bioconductor R package, seeking to classify individuals into two types of cancer tissue and was originally presented in Alon et al. (1999). For the former, methods were compared based on MSE and number of non-zero variables, while for the latter two, methods were compared according to accuracy, precision, area under the curve (AUC) and number of non-zero variables. For the two classification analyses, linear discriminant analysis (LDA) and SDSPCA were used instead of PLS and SPLS.

Table 4: Numerical Results with respective standard errors for Alon's Colon Cancer Dataset for $q = 2$ components.

| Method | Precision | Accuracy | AUC | Non-zero vars |
|--------|-----------|----------|-----|---------------|
| PCA | 0.6352 (0.0443) | 0.6000 (0.0441) | 0.5 (0.0) | 2000.0 (0.0) |
| LDA | 0.8308 (0.0480) | 0.7846 (0.0251) | 0.9 (0.0) | 2000.0 (0.0) |
| HSIC | 0.8380 (0.0508) | 0.7846 (0.0377) | 0.9 (0.0) | 2000.0 (0.0) |
| Bair | 0.6963 (0.0489) | 0.6538 (0.0417) | 0.7 (0.1) | 1342.9 (50.5) |
| CSPCA | 0.8380 (0.0508) | 0.7846 (0.0377) | 0.9 (0.0) | 2000.0 (0.0) |
| SDSPCA | 0.7973 (0.0965) | 0.6692 (0.0364) | 0.7 (0.1) | — |
| SPCA | 0.6563 (0.0415) | 0.6462 (0.0348) | 0.6 (0.0) | 13.5 (0.5) |
| SSPCA | 0.8394 (0.0571) | 0.8000 (0.0328) | 0.8 (0.0) | 225.3 (106.8) |
| **SCS-PCA** | 0.8421 (0.0446) | 0.8077 (0.0367) | 0.9 (0.0) | 275.2 (72.0) |

The mice dataset (Bushel et al., 2007) consists of $p = 3116$ genes and $n = 64$ male rats, exposed to varying doses of acetaminophen (paracetamol). The response variable used is levels of Albumin (ALB.g.dl). A $60\% - 20\% - 20\%$ split into training, validation — for hyperparameter tuning — and test sets was considered 10 different times and results were averaged across all splits. For hyperparameter tuning, cross-validation was used with MSE as optimisation metric. We consider the top $q = 2$ components for all projections. Table 3 illustrates the Monte Carlo estimates derived from the 10 splits. SCS-PCA achieves simultaneously the lowest MSE (0.6009) and strongest shrinking effect among all methods with a supervised objective — SPCA imposes sparsity on an unsupervised objective. PLS (0.6194) and SPLS (0.6978) follow in terms of prediction performance, with CSPCA also performing well (0.7249). SSPCA is the second most effective method in variable selection across supervised methods, however the prediction performance is not satisfactory (0.9285). PCA and SPCA are the worst performing methods in terms of prediction error, however, the latter has the strongest shrinking effect among all methods.

The leukemia dataset (Golub et al., 1999) comprises $p = 7129$ genes and $n = 72$ individuals. The dataset is already divided into training ($n_{tr} = 38$) and test ($n_{ts} = 34$) sets, seeking to classify patients into two types of leukemia, acute myeloid leukemia (AML) and acute lymphoblastic leukemia (ALL). No splitting is required for this dataset, however, 20% of the training data were used as validation set for tuning the hyperparameters where required. Results for accuracy, precision, AUC score and number of non-zero variables for $q = 2$ components are presented in barplots in Figure 1. SCS-PCA achieves the highest accuracy score (0.9706), while also providing the strongest shrinkage effect across supervised methods (15 non-zero variables). It achieves, along with SPCA using HSIC, CSPCA and SSPCA a perfect 1.0 score in both precision and AUC. SSPCA follows in both accuracy and sparsity performance (0.8824 and 36 non-zero variables). In general, all supervised methods, as well as PCA, were outperformed by their sparse variants in terms of accuracy. SPCA had 14 non-zero variables that contributed most in variance explained by the first two components, while for SDSPCA, recovery of non-zero variables is inapplicable since sparsity is imposed directly on the principal components and not the loadings. Overall, for the original split into training and test, used by Golub et al. (1999), SCS-PCA provides the strongest performance across all metrics.

The final dataset we employ in our analysis, was introduced by Alon et al. (1999) and it is publicly available on the R Bioconductor. It is an expression set consisting of $p = 2000$ genes and $n = 62$ samples, 40 of which are from tumors and 22 are from normal biopsies from healthy parts of patients' colons. A $60\% - 20\% - 20\%$ split into training, validation — for hyperparameter tuning — and test sets was considered 10 different times and results were averaged across all splits. For hyperparameter tuning, cross-validation was used with the logistic loss as optimisation metric. We used $q = 2$ components for all projections. Numerical results for accuracy, precision, AUC and number of non-zero variables — along with the corresponding standard errors — are presented in Table 4. SCS-PCA was the best performing method in terms of accuracy ($0.8421 \pm 0.0446$) and precision ($0.8077 \pm 0.0367$), with SSPCA closely behind ($0.8394 \pm 0.0571$ and $0.8000 \pm 0.0328$ respectively), with the former also having a higher AUC score over the latter (0.9 and 0.8 respectively). Strong performance in both precision and accuracy was also provided by their non-sparse counterparts, CSPCA and supervised PCA using HSIC, while PCA and SPCA were the poorest performing methods. SDSPCA and LDA also provided competitive results but still inferior to the top methods. In terms of shrinkage, SSPCA had a stronger effect than SCS-PCA (225 against 275 non-zero variables), although the standard error was higher

Figure 1: Results for Golub's Leukaemia Dataset for $q = 2$ components.

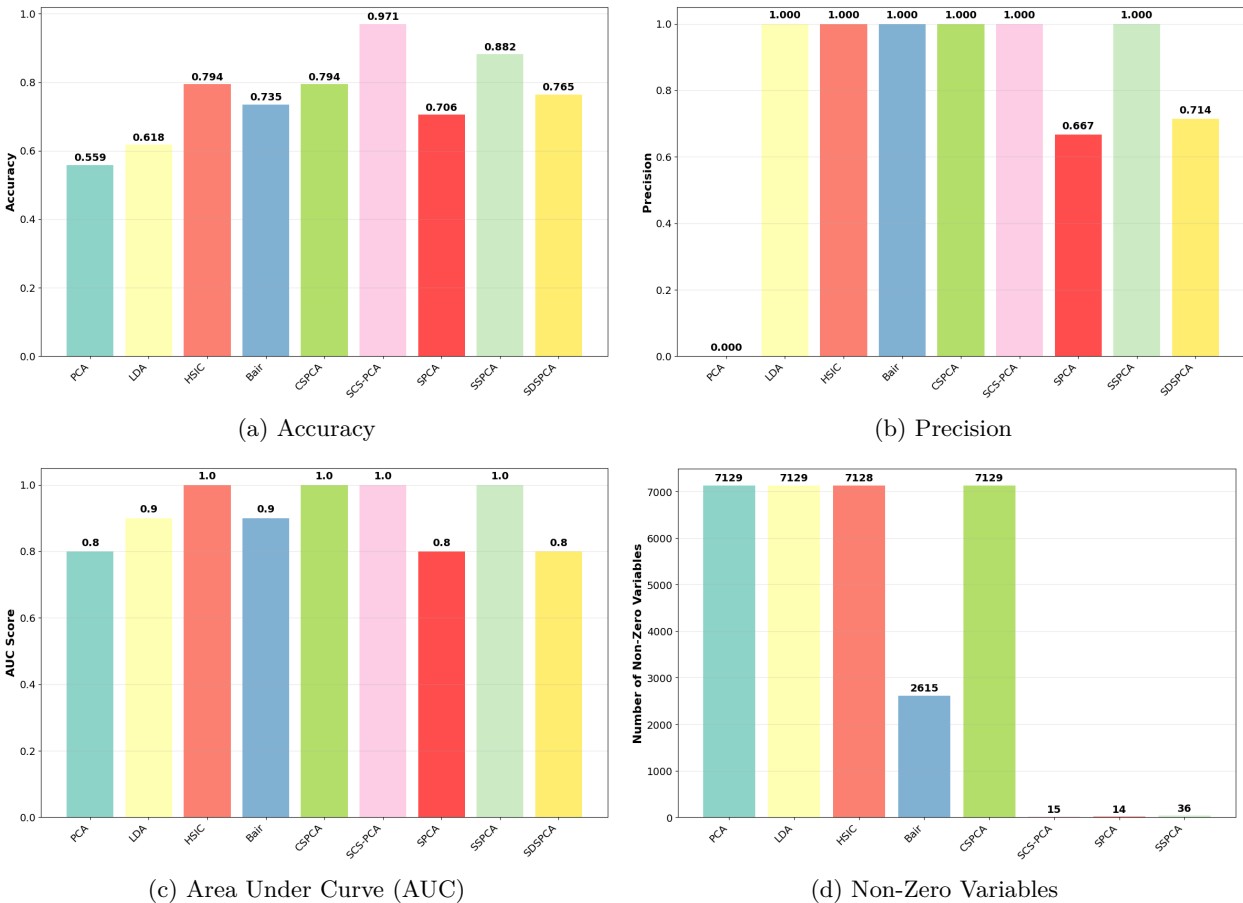

(a) Accuracy

(b) Precision

(c) Area Under Curve (AUC)

(d) Non-Zero Variables

(107 against 72 respectively). Overall, SCS-PCA provided strong predictive performance while also balancing well the sparsity effect, aiding in interpretability.

# 6 Discussion

In this paper, we have proposed a novel sparse supervised principal component analysis method, called sparse covariance supervised PCA (SCS-PCA), that incorporates sparsity into supervised PCA in the form of an $L_1$ penalty on the loadings of the projection matrix and performs simultaneous supervised dimensionality reduction and variable selection. We employ manifold proximal gradient descent to solve the proposed non-convex, non-smooth optimisation problem, while also providing the theoretical guarantees for global convergence to a stationary point under minimum requirements.

Our proposal seeks for a trade-off between response relevant and interpretable projections within a single objective function, while solving the non-convex optimisation problem directly. It is applicable to all types of response variables and offers guaranteed global convergence to a stationary point.

We have compared our method against existing sparse supervised PCA methods, supervised PCA methods, as well as baseline methods such as sparse PCA and sparse PLS for both regression and classification tasks. SCS-PCA offers competitive performance, improving prediction accuracy, while also enhancing interpretability due to the strong shrinkage effect it offers. Simulations and applications on real-world genetic datasets illustrate SCS-PCA's practical strength.

Our work fills a significant gap in sparse supervised PCA's literature, namely the direct trade-off between prediction accuracy and sparsity in a single objective function and the robust optimisation framework, along with the theoretical proofs of convergence. Some future directions include the extension of SCS-PCA to impose sparsity locally, or component-wise, since the current formulation applies sparsity globally on the loadings of the projection matrix. The framework presented here is deterministic, i.e. no probability distribution assumptions are made. A natural extension is to embed our method within a probabilistic framework, for example by placing sparse-inducing priors (e.g. spike-and-slab or Laplace) on the loadings of a supervised probabilistic PCA formulation (Yu et al., 2006). Such an approach would allow uncertainty quantification for the selected variables and may reduce the tuning cost of cross-validation, since the prior itself induces sparsity — a strategy already explored in the unsupervised setting by Ning & Ning (2024).

### Data Availability Statement

All data generating mechanisms used for the simulations, as well as the respective code used for the analysis are publicly available at the following GitHub repository `https://github.com/Anonymus-paper/SCSPCA`. The real-world datasets used during the experimental section are also publicly available. The liver toxicity data can be accessed in `https://www.rdocumentation.org/packages/mixOmics/versions/6.3.2/topics/liver.toxicity`, Golub's leukemia dataset is available on Kaggle `https://www.kaggle.com/datasets/crawford/gene-expression` and the colon cancer data can be acquired in `https://bioconductor.org/packages/release/data/experiment/html/colonCA.html`.

### Competing Interests

The authors have no competing interests to declare.

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

# A Appendix

## A.1 Theorem Proofs

*Proof of Theorem 1.* First, recall the definition of the proximal mapping associated to a convex function $f : \mathbb{R}^{n \times m} \to \mathbb{R} \cup \{+\infty\}$,

$$\operatorname{prox}_f(Y) := \arg\min_X f(X) + \frac{1}{2}\|X - Y\|_{\mathcal{F}}^2.$$

Let $h(X) = f(X) + \frac{1}{2}\|X - Y\|_{\mathcal{F}}^2$. Since $h$ is convex, as the sum of two convex functions, any minimizer is unique, which proves the uniqueness. There remains the proof of existence. Since $f$ is convex, it can be

lower bounded by an affine function, i.e. $f(X) \geq \langle X, A \rangle + b$, where $A \in \mathbb{R}^{n \times m}$, $b \in \mathbb{R}$ and $\langle X, A \rangle = \text{tr}(A^\top X)$ is the Frobenius inner product. Consequently,

$$h(X) \geq \langle X, A \rangle + b + \frac{1}{2}\|X - Y\|_{\mathcal{F}}^2 \geq \min_{X \in \mathbb{R}^n}\{\langle X, A \rangle + b + \frac{1}{2}\|X - Y\|_{\mathcal{F}}^2\} = c > -\infty,$$

making $h$ bounded below. We can easily show that all sublevel sets of $h$ are bounded since,

$$h(X) \leq \epsilon \implies \langle A, X \rangle + b + \frac{1}{2}\|X - Y\|_{\mathcal{F}}^2 \leq \epsilon \Leftrightarrow \|X - (Y - A)\|_{\mathcal{F}}^2 \leq \epsilon - b + \frac{1}{2}\|A\|_{\mathcal{F}}^2 = C,$$

where $C > 0$ is a constant. Thus, the sublevel set $\{X : h(X) \leq \epsilon\}$ is contained in a Frobenius norm ball and is therefore bounded. Now, let $(X_i)$ be a sequence such that $h(X_i) \downarrow \inf_X h(X)$. The sequence $(X_i)$ lies in the sublevel set $\{X : h(X) \leq h(X_1)\}$, which is closed and bounded. By the Bolzano-Weirstrass theorem, there exists a convergent subsequence $(X_{ik})$ such that $X_{ik} \to X^*$ and by the lower semicontinuity of $h$,

$$h(X^*) \leq \lim_{k \to \infty} \inf h(X_{ik}) = \inf h.$$

Thus $X^*$ is a minimiser of $h$, which is also unique, thus the proximal mapping is well-defined. □

*Proof of Theorem 2.* Provided the assumptions of Theorem 2 hold, the result follows directly from Theorem 5.5 of Chen et al. (2020).

**Condition (i).** The Euclidean gradient $\nabla f(W) = -2CW$ gives

$$\|\nabla f(W_1) - \nabla f(W_2)\|_{\mathcal{F}} = 2\|C(W_1 - W_2)\|_{\mathcal{F}} \leq 2\|C\|_{\text{op}}\|W_1 - W_2\|_{\mathcal{F}},$$

so $\nabla f$ is Lipschitz continuous with constant $L = 2\|C\|_{\text{op}} < \infty$, since $C$ is symmetric positive semi-definite.

**Condition (ii).** The function $h(W) = \eta\|W\|_1$ is a non-negative weighted norm, hence convex and Lipschitz continuous.

**Condition (iii).** The polar retraction equation 8 satisfies, for all $W \in \mathcal{S}(p, q)$ and $V \in \mathcal{T}_W \mathcal{S}(p, q)$,

$$\|\text{Retr}_W(V) - W\|_{\mathcal{F}} \leq M_1\|V\|_{\mathcal{F}},$$
$$\|\text{Retr}_W(V) - (W + V)\|_{\mathcal{F}} \leq M_2\|V\|_{\mathcal{F}}^2,$$

for finite constants $M_1, M_2 > 0$. Both properties hold on the compact Stiefel manifold (Absil et al., 2008).

**Condition (iv).** The step size and line-search parameters are set as specified in Algorithm 1.

With all four conditions verified, the proof proceeds via Lemmas 5.1–5.3 of Chen et al. (2020). By Lemma 5.1, the subproblem

$$V_k := \arg\min_{V \in \mathcal{T}_{W_k}\mathcal{S}} \langle \nabla f(W_k), V \rangle + \frac{1}{2t}\|V\|_{\mathcal{F}}^2 + \eta\|W_k + V\|_1$$

ensures sufficient decrease:

$$F(\text{Retr}_{W_k}(\alpha V_k)) \leq F(W_k) - \frac{\alpha}{2t}\|V_k\|_{\mathcal{F}}^2.$$

If $V_k = 0$, then $W_k$ satisfies the first-order optimality condition

$$0 \in \text{grad } f(W_k) + \text{Proj}_{\mathcal{T}_{W_k}\mathcal{S}} \, \partial h(W_k).$$

Since $C$ is positive semi-definite, $F$ is bounded below on the compact Stiefel manifold, so $\{F(W_k)\}$ is monotonically decreasing and bounded below. By Lemma 5.2,

$$\lim_{k \to \infty} \|V_k\|_{\mathcal{F}}^2 = 0,$$

and by Lemma 5.3, every accumulation point $W^*$ of $\{W_k\}$ is a stationary point. The conclusion follows from Theorem 5.5 of Chen et al. (2020). □

Table 5: Initialisation sensitivity for SCS-PCA. Random starts are summarised by mean, standard deviation and range across 100 runs.

| Initialisation | Test MSE | Non-zero | Objective |
|---|---|---|---|
| Random (mean $\pm$ s.d.) | $0.64 \pm 0.19$ | $2.3 \pm 0.5$ | $4379 \pm 877$ |
| Random (range) | $[0.19,\ 1.15]$ | $[2,\ 3]$ | $[2395,\ 5187]$ |
| CSPCA warm start | **0.38** | **3** | **3147** |

## A.2 Initialisation Sensitivity

To assess the practical impact of initialisation, we ran SCS-PCA from 100 random Stiefel initialisations and from the recommended CSPCA warm start on a single dataset from Simulation 1 ($n = 100$, $p = 500$, $q = 2$). Results are summarised in Table 5.

The CSPCA warm start achieves lower prediction error and recovers more true variables than the vast majority of random initialisations. While a small number of the 100 random starts reached comparable or even lower MSE (minimum 0.19), the average random start converges to a stationary point with MSE nearly twice as high as the CSPCA-initialised solution (0.64 vs 0.38) and selects fewer true variables (2.3 vs 3). These results support the recommendation in Section 4.5 to initialise with the CSPCA solution.

## A.3 Pareto Front Analysis

To examine the sparsity–prediction trade-off offered by each sparse supervised method, we evaluate SCS-PCA, SPLS and SSPCA across a range of penalty values on 100 independent datasets from Simulation 1 ($n = 100$, $p = 500$, $q = 2$, i.i.d. scenario). For each method, the penalty parameter is varied over a grid while all other settings remain fixed, and the average test MSE and number of selected variables are recorded. The resulting Pareto fronts are displayed in Figure 2.

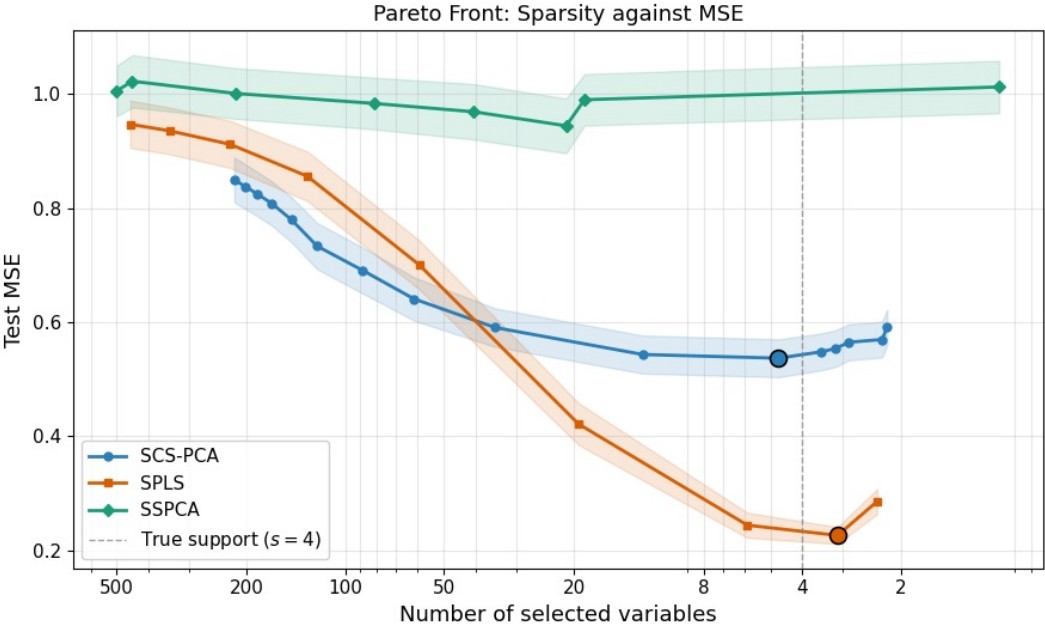

Figure 2: Pareto front for Simulation 1 (i.i.d., $q = 2$). Curves show the trade-off between test MSE and number of selected variables as the sparsity penalty varies. Shaded regions indicate $\pm 1$ standard error across 100 datasets. The dashed line marks the true support size ($s = 4$). Black-edged markers indicate each method's optimal MSE.

Both SCS-PCA and SPLS exhibit a clear elbow near the true support size, confirming that the sparsity penalty successfully drives variable selection toward the relevant features. SSPCA, by contrast, does not produce a meaningful trade-off: its MSE remains above 0.94 regardless of the constraint parameter.

SCS-PCA displays a smooth, monotonic descent as the penalty increases, reaching its minimum MSE (0.54) at approximately 5 non-zero variables. Notably, the curve remains stable in the neighbourhood of the true support size, with MSE varying by less than 0.05 between 3 and 8 selected variables. This suggests that the method is not overly sensitive to the exact choice of $\eta$ once the penalty is in the correct regime — a practically desirable property that reduces the burden of hyperparameter tuning.

SPLS achieves a lower minimum MSE (0.23) on this linear simulation, consistent with the main results in Table 1. However, its curve is steeper: MSE rises sharply on both sides of the optimum, indicating greater sensitivity to the penalty parameter. In the under-penalised regime, SPLS retains over 100 variables while SCS-PCA has already reduced to fewer than 35 at comparable MSE levels, reflecting the stronger shrinkage effect of the $\ell_1$ penalty on the projection matrix.

