# OpenReview forum: "Sparse Covariance Supervised Principal Component Analysis"
_TMLR — Under review for TMLR_

### Review · Reviewer_Ff8F · 2026-06-24

**Summary Of Contributions:**

The paper proposes a new method for supervised dimensionality reduction, termed Sparse Covariance Supervised PCA (SCS-PCA). The approach builds on covariance supervised PCA (CSPCA) and incorporates sparsity through an l1 penalty on the weight matrix, aiming to achieve a trade-off between predictive performance and interpretability. The resulting optimization problem is formulated over the Stiefel manifold and is solved using a manifold proximal gradient (ManPG) method. The authors also claim theoretical guarantees of convergence to a stationary point and demonstrate the performance of the method through simulations and applications to high-dimensional biological datasets.
Strengths:
The paper addresses an important problem, combining supervision and sparsity in a PCA-type framework.
The use of manifold optimization (ManPG) is appropriate for handling orthogonality constraints and is a natural extension of recent advances in optimization on manifolds.
The empirical evaluation includes both simulated and real-world high-dimensional datasets, which is relevant for assessing the practicality of the method.
Weaknesses:
The theoretical development lacks rigor in key places, particularly in the justification of the optimization framework (e.g., the role of convexity under manifold constraints and handling of the Riemannian gradient).
The experimental pipeline is not fully specified, especially regarding how predictions are obtained after dimensionality reduction, making comparisons difficult to assess for fairness and reproducibility.
The evaluation of variable selection performance is limited, as it only reports the number of selected variables without assessing recovery of relevant features.
Several empirical claims (e.g., consistent outperformance) are not supported by statistical validation.

**Additional Comments:**

For the authors: I encourage the authors to revise the manuscript with a stronger focus on clarity, rigor, and reproducibility, as this would significantly improve the contribution.

For the editor: The paper presents a potentially interesting idea but currently lacks sufficient rigor in the theoretical development and clarity in the experimental evaluation. The issues appear substantial but likely addressable with a thorough revision.

**Audience:**

Yes

**Audience Explanation:**

Combining supervised dimensionality reduction with sparsity under orthogonality constraints is relevant to the TMLR audience, particularly those working in high-dimensional statistics, machine learning, and optimization on manifolds. The proposed method and its connection to manifold proximal gradient techniques are likely of interest to researchers in these areas.

**Broader Impact Concerns:**

There are no immediate ethical concerns or direct societal risks associated with the proposed method.

**Claims And Evidence:**

No

**Claims Explanation:**

The paper’s claims are only partially supported by the presented evidence. Theoretical guarantees rely on the ManPG framework, but several key assumptions are not explicitly verified, and important steps (e.g., handling of the Riemannian gradient and convexity arguments) lack sufficient rigor.
Empirically, the experimental setup is not fully specified, particularly regarding the prediction pipeline and the use of validation versus cross-validation, which affects reproducibility and the fairness of comparisons. In addition, variable selection is evaluated only by the number of selected features, and no measure is provided to support the quality of the selected variables.
Overall, these issues limit the extent to which the evidence can be considered fully clear and convincing.

**Requested Changes:**

Clarify and strengthen the theoretical justification
1. Provide a complete and rigorous verification of the assumptions required for the ManPG framework (Chen et al., 2020), including smoothness, Lipschitz continuity, and properties of the retraction used.
2. Clarify the role of convexity in the presence of the non-convex Stiefel manifold constraint. The current discussion is misleading and should be corrected.
3.Explicitly derive the Riemannian gradient (including the projection onto the tangent space) rather than implicitly replacing it with the Euclidean gradient.
4. Ensure that convergence claims are stated precisely (i.e., convergence to stationary points rather than implying stronger guarantees).

Clarify the experimental pipeline
1.Clearly describe how predictions are obtained after dimensionality reduction.
2. Ensure that all competing methods are evaluated under comparable and fair conditions. (sparse, supervised, etc)
3.In simulations, report metrics that assess recovery of relevant variables (e.g., true positives, false positives, precision/recall), rather than only the number of non-zero coefficients.
4. Clarify whether observed differences across methods are statistically meaningful.

Improve presentation
1. Revise sections where arguments are currently informal or implicit, especially in the optimization and convergence analysis.
2. Improve the overall clarity of the experimental section to enhance reproducibility.

---

> ### Author Response · Authors · 2026-07-18
> **For all changes please see the revised uploaded manuscript.**
>
> We thank the reviewer Ff8F for their thorough and constructive evaluation.
> - Provide a complete and rigorous verification of the ManPG assumptions; clarify the role of convexity in the presence of the non-convex Stiefel manifold constraint; ensure that convergence claims are stated precisely.
>
> 1) Theorem 2 now lists all four ManPG conditions as explicit hypotheses: (i) smoothness of $f$ with Lipschitz constant $L = 2\|C\|_{\mathrm{op}}$ (ii) convexity and Lipschitz continuity of $h$ (iii) retraction boundedness (iv) step size conditions. The Appendix proof verifies each against Theorem 5.5 of Chen et al. (2020). We have also clarified in the introduction (first paragraph of page 3 that "global convergence" refers to convergence to a stationary point from an arbitrary initialisation, not to a global minimiser, following standard manifold optimisation terminology (Boumal et al. 2019).
> We corrected the misleading convexity language: $f(W) = -\mathrm{tr}(W^\top CW)$ is not convex
> but the ManPG framework requires only smoothness with Lipschitz gradient, not convexity of $f$. Section 4.3 now describes $f$ as "smooth," and the inapplicable $\mathcal{O}(1/k)$ rate claim has been removed from Section 4.5.
>
> - Explicitly derive the Riemannian gradient (including the projection onto the tangent space) rather than implicitly replacing it with the Euclidean gradient.
>
> 2) We have added a new block of material in Section 3 (eq. (5)--(7)) that: (a) states that we equip the Stiefel manifold with the embedded (Euclidean) metric citing Absil et al. (2008); (b) derives the Riemannian gradient as the orthogonal projection of the Euclidean gradient onto the tangent space, identifying the residual as lying in the normal space and (c) proves the equivalence $\langle \mathrm{grad}\, f(W), V \rangle = \langle \nabla f(W), V \rangle$ for all $V \in \mathcal{T}_W\mathcal{S}(p,q)$.
>
> - Clearly describe how predictions are obtained after dimensionality reduction; ensure that all competing methods are evaluated under comparable and fair conditions; report metrics that assess recovery of relevant variables (e.g., true positives, false positives, precision/recall); clarify whether observed differences across methods are statistically meaningful; improve the overall clarity of the experimental section to enhance reproducibility.
>
> 3) Section 5.1 has been reorganised from scratch. We explicitly describe the prediction pipeline used for all projection-based methods. The training data are projected via ($Z = XW$), ordinary least squares regression is fitted on the projected data, and predictions on the test set are obtained by projecting through the matrix W and applying the fitted model. For SPLS, predictions are obtained directly as ($\hat{Y} = X\hat{\beta}$). This prediction procedure is applied consistently across all methods.
> We specify that all methods now explicitly share the same training/validation/test split and the same data standardisation. Hyperparameters are selected by minimising validation MSE over tuning grids, which will be provided in the updated GitHub repository.
> We have also revised the simulation study by increasing the number of simulation replicates from 20 to 100, reporting precision and recall alongside the number of selected variables, and statistical significance results using Wilcoxon signed-rank tests (the updated tables denote statistically significant advantages and disadvantages of SCS-PCA using the symbols ($^{\dagger}$) and ($^{\ddagger}$), respectively.). In addition, our code computes the numbers of true positives (TP), false positives (FP), and false negatives (FN) relative to the known true support ($\{X_1, X_2, X_3, X_4\}$). To improve readability and accommodate the additional metrics, Simulation 3 has been removed, as both Simulations 2 and 3 considered nonlinear response relationships. Updated tables are available on Section 5.1.
> We believe these additions, together with the publicly available code repository (which will be uploaded later due to time constraints), substantially improve the reproducibility and clarity of our work.
>
> - Revise sections where arguments are currently informal or implicit, especially in the optimization and convergence analysis.
>
> 4) The following changes have been made:
> - Theorem 2 and its proof have been formalised
> - We specify that the polar retraction is used and give its explicit formula (eq 8)
> - A new paragraph in Section 4.5 details the per-iteration computational complexity of Algorithm 1, identifying the $O(p^2 q)$ gradient computation as the dominant cost and
> - The initialisation discussion in the Appendix now explains that while Theorem 2 guarantees convergence from arbitrary initialisation, we recommend CSPCA warm start to avoid inferior stationary points, which is supported by a new initialisation sensitivity experiment.

---

### Review · Reviewer_VcSc · 2026-07-07

**Summary Of Contributions:**

This paper proposes a methodology for sparse supervised principal component analysis—a variant of PCA ensuring sparse and response-informative loadings. A novel criterion is introduced, making a tradeoff between accuracy and sparsity (via a classical $L_1$ penalty). It is maximized with a manifold proximal gradient algorithm on the Stiefel manifold—the space of $p \times q$ semi-orthogonal matrices, representing the components. Extensive synthetic and real-data experiments are performed and show the superiority of the proposed method over a large ensemble of natural competitors (including PCA, sparse PCA, supervised PCA, sparse supervised PCA), in terms of prediction accuracy and sparsity.

Overall I believe that the paper is a good contribution to the field of machine learning (ML). I was not aware of sparse supervised PCA methods before, but I believe that they have a well-justified practical interest—although (what appears to be) the founding paper, by **Sharifzadeh et al. (2017)**, is not so cited for an ML paper (59 citations according to Google Scholar), questioning its impact in the ML community. The paper is well written and structured. The contributions relying on Riemannian geometry are well exposed for an ML reader that might not be aware of the field. The theoretical and experimental results are convincing.

The paper could be strengthened in terms of claim justifications, theorem statements and presentation of geometric tools (see below for more details), but it is already in a good form.

---
References:
- Sara Sharifzadeh, Ali Ghodsi, Line H. Clemmensen, and Bjarne K. Ersbøll. Sparse supervised principal component analysis (sspca) for dimension reduction and variable selection. Engineering Applications of Artificial Intelligence, 65:168–177, 2017.

**Additional Comments:**

- The paper seems to be a sparse follow-up of **Papazoglou and Yin (2025)** by the same authors (the github link in the conclusion did betray anonymity). I do not believe that this is a problem, as the differences are clear (the former does not produce sparse loadings) and do not overlap, but I just wanted to precise it somewhere in case.
- Since there are many minor requested changes, could the authors please return a "track-changes" version of the next submission to ease review?

---
References:
- Theodosios Papazoglou and Guosheng Yin. Covariance supervised principal component analysis. arXiv, 2025. URL https://arxiv.org/abs/2506.19247.

**Audience:**

Yes

**Audience Explanation:**

I think so: sparse supervised PCA seems like a general-purpose machine learning problem without much related work. This paper makes a significant advance in that problem by proposing a new criterion (aligning with classical sparse methods) and directly optimizing it via Riemannian proximal gradient descent (consequently enjoying convergence guarantees) instead of resorting to auxiliary functions and alternating algorithms like previous literature. The empirical results are convincing, ranking the proposed method in the first place among a wide range of PCA and variant methods in terms of prediction accuracy and sparsity. The code—available on GitHub—is quite generic, making it useable on a wide range of problems.

**Broader Impact Concerns:**

A general sentence regarding the ethical impacts of ML in general could be added, but not mandatory.

**Claims And Evidence:**

Yes

**Claims Explanation:**

The most important claims—theoretical analysis and practical competitiveness with baseline methods—are well evidenced by the proofs and numerical experiments.

A theoretical motivation for the proposed algorithm is that known SSPCA algorithms have "no robust convergence guarantees and are sensitive to initialisation" (claimed in the abstract). Although partially discussed, notably in last paragraph of page 2, I believe the justification for this claim is insufficient: it should be clarified. I notably checked the work by **Feng et al. (2019)** and they do include a convergence analysis in their paper, so the authors should maybe (at least) compare their convergence guarantees to Feng et al.'s and other mentioned works. Moreover, while sensitivity to initialisation is mentioned as a limit of previous literature (without much justification), it is not really addressed in this paper either (the authors recommend initialising via the solution of the unpenalized problem without further insight).

The argument of "interpretability" to motivate sparse methods is often used. While I clearly understand that sparse components improve interpretability—as they express as combinations of a few variables—and I am not expecting authors of a methodological ML paper to dive deeply into the applications, I believe that justifying the interest of sparsity by actually trying to interpret the resulting components would enrich the claim that sparse methods are relevant for such a problem. But this is optional if this is out of the authors' area of expertise.

---
References:
- Chun-Mei Feng, Yong Xu, Jin-Xing Liu, Ying-Lian Gao, and Chun-Hou Zheng. Supervised discriminative sparse pca for com-characteristic gene selection and tumor classification on multiview biological data. IEEE Transactions on Neural Networks and Learning Systems, 30(10):2926–2937, 2019.

**Requested Changes:**

- Major
	- Better nuance the claims (cf. remarks in the previous frame) or better justify them.
	- Theorems 1 and 2 should be stated more rigorously (notably, taking back the hypotheses).
	- I believe the authors forgot some traces or norms in some of the optimization problems, for instance in (1), (2), (3): the criterion is not scalar... I guess this is just a typo.
	- The authors do not talk about Riemannian metric in the paper, which is fundamental to define the Riemannian gradient. They should at least define that before introducing the gradient, and precise which metric they use (both "embedded" and "canonical" metrics are classical for the Stiefel manifold), because the notation $⟨grad_f (x), v⟩_x$ depends on the metric. See, for instance, Section 3.6 of **Absil et al. (2008)**.
	- The definition of a retraction is probably not the right one. See Definition 4.1.1 of **Absil et al. (2008)** for the one that is usually considered in optimization algorithms on manifolds to prove convergence. Moreover, the authors do not precise which retraction is considered in the algorithm (they should give a precise expression) and the equation at the bottom of page 8 does not seem to match the definition 5 in terms of input and output spaces.
	- I think it could be interesting to discuss the invariances of the considered problems. For instance, PCA ($\min_{W \in \mathbb{R}^{p \times q}\colon W^\top W =I_q} \|X - XW W^\top\|_F^2$) is invariant to rotation (right multiplication of $W$ by an orthogonal matrix), meaning that any basis of the principal subspace works. Sparse PCA involves additional norms on the matrix, leading to a loss of the rotational invariance, but the criterion is still invariant to permutation of the columns of  $W$, or to sign switch for instance. These invariances influence the optimization landscape and might lead to issues in the optimization algorithm (see, for instance, **Absil et al. (2008)**, Section 2.1.1).
- Minor
	- Abstract
		- "is able to select more features": isn't it rather "less features" if you are doing sparsity?
	- Section 1
		- "It projects the original high dimensional data into a lower dimensional, orthogonal subspace": the word "orthogonal" is inappropriate I think (with respect to what?). I would replace with "linear".
		- "As far as we are concerned": replace with "to the best of our knowledge"?
		- "Since the two objectives are competing, the two objectives are balanced via": repetition -> "they are balanced via"
		- "rather as" -> "rather than as"
	- Section 2
		- "etc.." -> "etc."
		- "RKHs" -> "RKHS"
		- "CSPCA maintains data covariance structure, matches, approximately, PCA in variance explained, and outperforms" -> "CSPCA maintains data covariance structure, approximately matches PCA in explained variance, and outperforms"
		- "From (4) we can observe how sparsity (β) is imposed": the corresponding equation is not labeled
	- Section 3
		- "differential manifold" -> "differentiable manifold"
		- "e.g. LASSO regression": add the reference
		- The space $X$ is not defined in definition 5 and it overlaps with the notation for the dataset $X$. Anyway, I believe that the definition of retraction should be modified (see above remark).
		- "lower- semicontinuous" -> "lower semicontinuous"
		- Is Theorem 1 an original contribution? If not, cite the original reference or a book in the theorem name.
	- Section 4
		- "rather a constrained" -> "rather than a constrained"?
		- "Unlike existent methods" -> "Unlike existing methods"?
		- "Since the objective function in (5) is non-differentiable, proximal gradient descent algorithms are required": This sentence gives the impression that there are no other Riemannian optimization algorithms to deal with nondifferentiable functions.
		- "Caley" -> "Cayley"
		- "In the case of the L1 norm, the proximal mapping is defined as the soft-thresholding operator": add reference
		- Algorithm: "Require: Initial $W_0 \in \mathcal M$": I would add, maybe in parenthesis, "CSPCA" or "random" to clarify "initial".
		- "if and only if both parts of the objective function are convex and additionally the gradient of the smooth part": replace "parts" with "terms"?
		- "In practice, we suggest to initialise Algorithm 1 using the solution to the standard CSPCA": Maybe include a larger discussion about initialisation, saying for instance that we could initialise randomly (via polar decomposition of Gaussian matrix, see Theorem 1.5.5. of **Chikuse (2003)**) but this may converge to local minima. If you have simulated results testifying the superiority of initialising with CSPCA, then referring to them would strengthen the claim.
		- "The computational burden of the algorithm lies in solving the subproblem to obtain Vk, while the RSSN method for finding the descent direction performs robustly and efficiently (Chen et al., 2020).": add the equation reference after "subproblem". Moreover, giving the computational complexities of each step would strengthen the paper.
		- "CSPCA is not very sensitive to the choice of κ, and large fluctuations are needed to observe a significant difference.": I guess this is a heuristic argument, so please add something like "we observe in practice that...", because this belongs to the theoretical section.
	- Section 5
		- The tables are pretty loaded. You could maybe remove a few significative digits and multiply the MSE (and standard error) by 10 for readability.
		- Equation for simulation 2: shouldn't it be "X_4 + \epsilon"?
		- "All simulations were performed for q = 2, 3 and 4 components": I guess you truncate the simulation equations at the $q$-th term (and add noise)? Maybe precise that because this is not evident at first sight.
		- The tables are not all well positioned with respect to the text.
		- Could you explain why SCS-PCA performs in general better than SSPCA? Preferably by a technical analysis of the optimization algorithms, otherwise heuristically.
		- The use of bold font is not consistent across tables. The names of the columns neither: "MSE" vs "MSE (s.e.)", "Non-zero variables" vs "Non-zero", etc.
		- Table 5: put more significative digits for AUC.
		- Why making a figure for the Leukemia dataset and not a table like the other ones? Moreover, contrary to what is stated, the standard errors are not displayed in the figure. You could add $1 \sigma$ error bars.
		- Can an interpretation be made out of the 15 retained components for the Leukemia dataset? How does it compare to the retained components of SPCA and SSPCA? Does it match historical findings? This is optional but it would really strengthen the paper.
	- Section 6
		- "state-of-the-art baselines such as sparse PCA and sparse PLS": I would rather say "baseline methods such as sparse PCA and sparse PLS", because these are pretty standard methods.
		- "This extension may also aid in reducing the tuning cost created from cross-validation through a prior distribution that will naturally perform sparsity.": the prior distribution might itself rely on a hyperparameter to control sparsity (see, for instance, Bayesian PCA from Bishop), so please clarify the sentence.
		- The perspectives should be accompanied with references for each idea, because these can be quite readily implemented via classical existing works.
	- Code
		- The code repository should be better presented, with .py files for reproducibility. Also detail in the "Data Availability Statement" how the other methods are implemented, based on which code / package, acknowledging the open-source code you used for competitors (ex: scikit-learn or authors' implementations repositories).
	- Bibliography
		- Replace "\citet{a, b}" with "\citet{a} and \citet{ b}" for multiple citations. There are also some citations that should be with parentheses (\citep), notably in Section 5.
		- The bibliography needs a complete rework, entry by entry. I noticed several issues ("Ian T Jolliffe. Introduction, pp. 1–9. Springer New York", "Neal Parikh, Stephen P. Boyd, and publisher. Now Publishers. Proximal algorithms. Foundations and trends in optimization", etc.), as well as inconsistencies (some journals abbreviated and other not, missing months, URLs, etc.). See, for instance, https://jmlr.org/format/formatting-errors.html for common formatting errors.
	- Appendix
		- I find it strange to have a unique section named "Appendix".
		- The hyperlinks do not seem to work in appendix, and some in the main text are broken too. Make sure you are using the correct journal template without extra conflicting packages and you are compiling correctly.
	- Other
		- There are words in American English convention (e.g., optimization) and other ones in British English (e.g., generalisation). Please homogenize.
		- Use \operatorname or \mathrm for trace, grad, prox, etc.
		- I believe $\mathrm{St}(p, q)$ is a more common notation than $S(p, q)$ for the Stiefel manifold
		- I do not remember seeing a discussion on the choice of the intrinsic dimension $q$. Do you have any clues?


---
References:
- Bishop, C. (1998) « Bayesian PCA », _Advances in Neural Information Processing Systems_. https://papers.nips.cc/paper/1998/hash/c88d8d0a6097754525e02c2246d8d27f-Abstract.html
- Absil, P.-A., Mahony, R. et Sepulchre, R. (2008) _Optimization Algorithms on Matrix Manifolds_. Princeton, NJ: Princeton University Press. https://www.degruyter.com/document/doi/10.1515/9781400830244/html
- Chikuse, Y. (2003) _Statistics on Special Manifolds_. New York, NY: Springer (LNS). https://doi.org/10.1007/978-0-387-21540-2

---

> ### Author Response · Authors · 2026-07-18
> **We sincerely thank the reviewer VcSc for their exceptionally thorough and detailed evaluation.**
>
> - Minor corrections: typos, notation, language, formatting, references, and presentation issues throughout the manuscript.
>
> 1) All minor corrections have been addressed in the revised manuscript and are highlighted in red. These include typographical corrections (including the traces in equations 1-3), consistent British English usage, notation standardisation, equation references, bibliography formatting, citation style, hyperlink verification, and formatting consistency. We also revised ambiguous statements, corrected table formatting, ensured consistent use of mathematical operators, and improved the overall presentation throughout the manuscript.
>
> - Better nuance the claims regarding convergence guarantees and initialisation.
>
> 2) We have revised the introduction to better nuance our claims. We now acknowledge that some existing methods include convergence analyses (Feng et al., 2019), but clarify that these guarantees typically concern alternating iterations or surrogate formulations and do not establish convergence to a stationary point of the original penalised problem. We also clarify that ``global convergence'' refers to convergence to a stationary point from arbitrary initialisation, following standard manifold optimisation terminology (Boumal et al., 2019), rather than convergence to a global minimiser.
> Section 4.5 now discusses initialisation: although Theorem 2 guarantees convergence from any point on the Stiefel manifold, different initialisations may lead to different stationary points. We recommend the CSPCA warm start and provide a detailed sensitivity analysis in Appendix A2 over 100 random Stiefel initialisations.
>
> - Theorems 1 and 2 should be stated more rigorously.
>
> 3) Both theorems have been rewritten as self-contained results. Theorem 1 now includes all required assumptions directly in the statement and cites Parikh & Boyd (2014). Theorem 2 now explicitly states the required conditions regarding smoothness, Lipschitz continuity, convexity, retraction properties, and step-size/line-search assumptions, together with a precise stationarity conclusion. The proof has also been reorganised to verify each assumption against Theorem 5.5 of Chen et al. (2020).
>
> - The Riemannian metric should be defined before introducing the Riemannian gradient.
>
> 4) We have added a new block in Section 3 defining the Riemannian metric on the Stiefel manifold. We specify the use of the embedded Euclidean metric, derive the Riemannian gradient as the projection of the Euclidean gradient onto the tangent space, define the normal space, and establish the equivalence between the two gradients. Section 4.3 now references this derivation when constructing the optimisation subproblem.
>
> - The definition of retraction should be corrected and the specific retraction used in the algorithm should be stated.
>
> 5) Definition 5 has been completely rewritten following Definition 4.1.1 of Absil et al. (2008). We explicitly specify the use of the polar retraction in eq. (8). The input and output spaces are also corrected to be consistent with the retraction step in Algorithm 1.
>
> - Discuss the invariances of the optimisation problem and their influence on the landscape.
>
> 6) We acknowledge this is an interesting point that could provide additional insight into the optimisation landscape. We note that the $\ell_1$ penalty breaks rotational invariance while column permutation and sign-flip symmetries are preserved, which may lead to equivalent stationary points. A formal treatment of these symmetries and their interaction with the ManPG landscape is beyond the scope of the current rebuttal due to time constraint, but we plan to include a dedicated discussion in the final revised manuscript.
>
> - The abstract should clarify that sparsity selects fewer variables.
>
> 7) The statement has been revised accordingly. It now states that SCS-PCA selects fewer features while outperforming projection-based methods in variable selection accuracy.
>
> - The experimental section requires clearer and more comprehensive evaluation; the repository should provide reproducible scripts and details of competing implementations.
>
> 8) The experimental section has been substantially revised. We now explicitly describe the prediction pipeline, standardisation procedure, and evaluation framework shared by all methods. The number of simulation replicates has been increased from 20 to 100, tuning grids have been expanded, and variable selection metrics including precision, recall and paired Wilcoxon signed-rank tests have been added to assess statistical significance of MSE differences.
> To improve clarity, Simulation 3 was removed. Tables have been reformatted for consistency, equations have been corrected, and additional discussion has been added to explain why SCS-PCA improves upon SSPCA. The code repository is being updated with standalone Python scripts and implementation details for all competing methods to improve reproducibility.

---

> > ### Comment · Reviewer_VcSc · 2026-07-21
> > **Second round of review**
> >
> > I would like to thank and congratulate the authors for their thorough work in such a small amount of time. I am overall happy with the modifications on my side. I also looked at the other reviewers' comments and I am also satisfied with the authors' answers. Perhaps the significance concern raised by reviewer `pYBu` is worth commenting. I agree that the contribution is rather incremental: it is a sparse extension of CSPCA, relying on a well-established non-smooth optimization algorithm on the Stiefel manifold. However, I believe that the paper is still of practical interest to the general machine learning audience, and the contributions are sound and particularly well exposed. Hence I believe it is suitable for TMLR. I still have a few comments:
> > - Major:
> > 	- Regarding the abstract claim "no robust convergence guarantees and are sensitive to initialisation": while I am now happy with your justification on the convergence guarantees (top of page 3), I am still not very convinced by the claim about initialisation: perhaps you should add some lines in Table 5 with the mentioned methods (SSPCA, SDSPCA, etc.) to justify their sensitivity with respect to initialisation (or remove this claim). Moreover, while SDSPCA seem to initialise randomly, SDSPCAAN by  Shi et al. seem to initialise in a deterministic way so I don't see why it would be sensitive to initialisation.
> > - Minor:
> > 	- "A closely related situation arises in statistical genetics, where fine-mapping analyses seek to isolate the few putatively causal variants driving an association signal from many correlated variants in linkage disequilibrium at a locus (Wu et al., 2026).": the sentence is not very understandable for the non-specialist.
> > 	- "The difficulty in obtaining rigorous optimisation schemes for sparse methods stems from the non-convexity of the feasible set, i.e. the orthogonality constraint, rather than the objective itself." -> I think there is a problem with this sentence. Because this is only later in this paragraph that you mention the transition from Euclidean sets to the Stiefel manifold. You probably mismatched two sentences in this paragraph.
> > 	- Consistency: a few $L_1$ are remaining: use $\ell_1$ everywhere
> > 	- Add citation for PLS (both when first mentioned in intro and when detailed in 2.4)
> > 	- The notation $S(p, q)$ for Stiefel manifold is put in red but has not been changed to $\mathrm{St}(p, q)$.
> > 	- "Throughout, we equip the Stiefel manifold": "Throughout the paper"?
> > 	- "so whenever the second argument is tangent": unclear. I guess you mean that when the Euclidean gradient belongs to the tangent space, it is equal to the Riemannian gradient (under the embedded metric)?
> > 	- Definition 5: replace point in (i) with comma and there is a repetition of the word "satisfying" for point (ii). Just reformulate it as in (i): "D RetrW (0W) = idTW M, where RetrW (0W) denotes the differential of RetrW at 0W"
> > 	- When introducing the expression for the polar retraction, you could cite Example 4.1.3 of Absil et al.'s book.
> > 	- Page 9: "By(7)": missing space
> > 	- "sparse-inducing priors": "sparsity-inducing priors"
> > 	- You need to do another pass on the bibliography: I spotted again several issues: "Pierre A. Absil vs. P-A Absil", "Robert. Mahony, and R. Sepulchre". Either put the full first names or abbreviate all of them for consistency.
> > 	- Table 5: perhaps merge lines 1 and 2 with a format "mean \pm s.d. (range)"? Because the second line looks odd at first sight: we could think you are comparing two initialisations.
> >
> >
> > PS: I will be off from Friday 24 July afternoon to Monday 17 August morning so I will take back the discussion at that time, unless you have specific points you want to discuss before this Friday.

---

### Review · Reviewer_pYBu · 2026-07-12

**Summary Of Contributions:**

The paper proposes a sparse supervised PCA variant deemed sparse covariance supervised PCA (SCS-PCA). Notably, the SCS-PCA relies on manifold optimization using the manifold proximal gradient descent (ManPG) procedure of (Chen et al., 2020). Specifically the approach optimizes the standard supervised PCA objective trace(W^T CW) where C = X^⊤YY^⊤X+\kappa X^⊤X, with W residing on the Stiefel manifold imposing conventional sparsity promoting l1 regularization on W. Global convergence to a stationary point is theoretically derived following Lipchitz continuity of the gradient without the (convex) L1 regularization. The approach is compared to relevant alternatives including existing supervised and sparse supervised PCA approaches as well as PLS methods and found to perform well although the results do not appear in general significant given the standard deviation. As such, the results are not overly convincing but the approach is valid and useful.

**Audience:**

No

**Audience Explanation:**

The approach is in my view rather incremental in implementing sparse supervised PCA using standard manifold optimization (i.e., ManPG) whereas the results are not overly convincing and it is thus unclear that the proposed methodology provide clear merits over existing approaches apart from a more principled optimization than conventional sparse supervised PCA. The approach as I see it is a rather straightforward extension of existing procedures by:

i)	Imposing sparsity as imposed in in Sparse supervised PCA by Sharifzadeh et al. (2017) using as kernel K=YY^T+\kappa I which is a trivial choice of kernel as I see it that corresponds to the CSPCA formalism.

ii)	Use the (ManPG) of (Chen et al., 2020) to properly optimize W on the Stiefel manifold, however,  ManPG has previously been used for sparse PCA in the work of Chen and its adaptation to the sparse supervised PCA seems in this context straightforward.

Consequently, the nature of this contribution appears in my view very incremental and the global convergence to a stationarity point results given in the Theorem also appears trivial and as a natural result of convergence guarantees for Sparse PCA using ManPG (correct me if I am wrong).

As such my concern is the nature of the contribution being very limited to garner interest and why I am here inclined to answer no.

Had the results been convincingly demonstrating that the new optimization improves substantially upon performance this would give the paper impact, but I find the results and the merits of the approach when compared to existing methodologies not overly convincing also given the reported error bars.

**Broader Impact Concerns:**

There are no immediate broader impact concerns of the developed methodology.

**Claims And Evidence:**

Yes

**Claims Explanation:**

The claims are generally sound and the paper clear and easy to read and overall the presentation is fine. The paper is thus overall well written and the approach presented in an OK manner. My concern here is the rather incremental nature of the contribution with the results also not being overly convincing, see interest.

The paper claims that the proposed approach does not rely on specifying a kernel, i.e. as stated
“our method can be applied for any type of response variable and does not require kernel tuning.”, but implicitly it is as I see it specifying the kernel K= YY^+\kappa I (that also depends on the hyperparameter \kappa) - this should be clarified.

**Requested Changes:**

The description of the PLS framework and sparse PLS is not very self-contained and needs to be elaborated upon. How is the standard PLS solved and for the sparse PLS variant how are multiple components solved for? The objective presented is only for a single component.

It is unclear how the alternative sparse methods are tuned and as such the sparsity penalty choice will heavily influence how many non-zeros elements are found compared to MSE performance. It would thus be better to benchmark the procedures in terms of a Pareto front of sparsity vs. MSE to gauge their dynamics in performance and use of few elements in W. This could be achieved simply by solving for the solution path across a series of l1 regularization parameters for the sparse approaches.

The paper also needs to better motivate and position the use of sparsity. As a regularization strategy to promote improved generalization other choices such as L2 regularization could also be considered. Typically, sparsity is used to promote easy explainable presentations using few aspects for the representation. However, the paper does not consider the explainability aspect of the learned sparse representations. The manuscript would thus benefit from clearly motivating the need for sparse representation as opposed to other types of standard regularization (i.e., using few components or regularizing by alternative norms than the l1).

The proposed manifold optimization is non-convex. Error bars should thus also be included in the histogram plots in Figure 1 across random initialization to reflect the potential issues of local minima in the optimization even though the split here is fixed.

---

> ### Author Response · Authors · 2026-07-18
> **For all changes please see the revised uploaded manuscript.**
>
> We thank the reviewer pYBu for their evaluation and for the concrete suggestions.
>
> - The paper claims that the proposed approach does not rely on specifying a kernel, i.e. "our method can be applied for any type of response variable and does not require kernel tuning," but implicitly it is specifying the kernel $K = YY^\top + \kappa I$ --- this should be clarified.
>
> 1) Section 4.1 has been revised to remove the blanket claim about kernel tuning. We now explain that SCS-PCA incorporates the response through the data-response covariance $X^\top YY^\top X$ directly, avoiding the need to select a kernel family and associated bandwidth parameters (as required by HSIC-based methods such as SSPCA). We note that $\kappa$ controls only the relative weight of the supervised and unsupervised components rather than the shape of a kernel function.
>
> - The description of the PLS framework and sparse PLS is not very self-contained. How is the standard PLS solved and for the sparse PLS variant how are multiple components solved for?
>
> 2) Section 2.4 has been expanded and details how the components to both PLS and sparse PLS are derived via deflation.
>
> - It is unclear how the alternative sparse methods are tuned and as such the sparsity penalty choice will heavily influence how many non-zero elements are found compared to MSE performance. It would thus be better to benchmark the procedures in terms of a Pareto front of sparsity vs. MSE.
>
> 3) The revised Section 5.1 now describes the hyperparameter tuning procedure for every method, while we also update the GitHub repository to include complete details on the grids used. We compute sparsity-vs-MSE Pareto fronts for all sparse supervised methods by evaluating each method across a range of penalty values on $100$ independent dataset replicates and include a dedicated subsection in the Appendix. We have reorganised the simulation framework entirely to improve clarity and communicate the strength of SCS-PCA clearly. Specifically, we have increased the number of independent replicates from $20$ to $100$. The revised evaluation now includes precision and recall and paired Wilcoxon signed-rank test results comparing the MSE of SCS-PCA against each competing method across the dataset replicates. To ease readability we have removed the final simulation scenario.
>
> - The paper needs to better motivate and position the use of sparsity. As a regularization strategy to promote improved generalization other choices such as $L_2$ regularization could also be considered.
>
> 4) We have added a new paragraph to the introduction addressing this point. Specifically, we explain that $\ell_2$  regularisation shrinks loadings but does not produce exact zeros, while the $\ell_0$ penalty is combinatorial and computationally intractable. The $\ell_1$ norm provides the tightest convex surrogate to $\ell_0$, admits a closed-form soft-thresholding proximal operator, and is therefore both the standard choice in the literature and well suited to our manifold proximal optimisation framework.
>
> - The proposed manifold optimization is non-convex. Error bars should thus also be included in the histogram plots in Figure 1 across random initialization to reflect the potential issues of local minima.
>
> 5) We are addressing this in two ways. First, we are converting Figure 1 (leukemia results) to a table format with standard errors, consistent with the other datasets. Due to time constraints, we will upload results on the next revision. Second, we have added a new initialisation sensitivity experiment (Appendix A2) that runs SCS-PCA from multiple random Stiefel initialisations on the same dataset and reports the distribution of objective values, test MSE, and number of non-zero variables.
>
> - Regarding the incrementality of the contribution and whether the results convincingly demonstrate that the new optimization improves substantially upon performance.
>
> 6) We respectfully note that our contribution extends beyond combining CSPCA with ManPG. Specifically, (a) we introduce a single objective function that directly balances prediction accuracy and sparsity through a tunable penalty, whereas SSPCA enforces sparsity via a constraint and solves an approximate reformulation (b) the revised theoretical development (Riemannian metric, gradient derivation, retraction specification, and self-contained theorem statements) provides a rigorous optimisation framework absent from the sparse supervised PCA literature and (c) the expanded experimental evaluation—including larger tuning grids, variable selection metrics (precision, recall), paired significance tests, and Pareto fronts—provides substantially stronger empirical evidence and demonstrates the practical advantages of SCS-PCA over SSPCA. We believe these revisions clearly illustrate that SCS-PCA offers both a more principled formulation and competitive empirical performance relative to existing approaches.